# Maternal perceptions of father involvement among refugee and disadvantaged families in Beirut, Lebanon

**Sascha Hein**[1]*, **Johanna Bick**[2], **Ghassan Issa**[3], **Lara Aoude**[3], **Cosette Maalouf**[3],
**Abir Awar**[3], **Sawsan Nourallah**[3], **Anna L. Zonderman**[4], **Sarah Householder**[5],
**Liliya Katsovich**[5], **Kaveh Khoshnood**[5], **Christina Moore**[6], **Rima Salah**[5], **Pia R. Britto**[7],
**James F. Leckman**[5], **Liliana Angelica Ponguta**[5]*

1 Department of Education and Psychology, Freie Universität Berlin, Berlin, Germany, 2 Department of
Psychology, University of Houston, Houston, Texas, United States of America, 3 Arab Resource Collective,
Beirut, Lebanon, 4 UChicago Medicine, The University of Chicago, Chicago, Illinois, United States of
America, 5 Child Study Center, Yale University, New Haven, Connecticut, United States of America,
6 Department of Psychology, University of Delaware, Newark, Delaware, United States of America,
7 UNICEF, New York, New York, United States of America

* Sascha.Hein@fu-berlin.de (SH); Angelica.Ponguta@yale.edu (LAP)

journal.pone.0229670

GERMANY

**Data Availability Statement:** All relevant data are
within the manuscript and its Supporting
Information files.

## Abstract

The role of fathers in (co-)parenting their children among refugee and disadvantaged families in low and middle-income countries (LMICs) remains poorly understood. This study sought to examine the associations among mothers' perceptions of their husband's involvement (hereafter referred to as paternal involvement), and her perceptions of her own well-being and a number of other variables, as well as observed mother-child interactions in families living in refugee and disadvantaged communities in Beirut, Lebanon. We analyzed baseline data from 104 mother-child dyads (mean age of children = 4.34 years; range = 2.05 to 7.93 years of age) who participated in a randomized controlled trial aimed at evaluating the impact of the Mother-Child Education Program in Beirut. In addition to the mother's perception of paternal involvement and the videotaped mother-child interactions, data were collected concerning the mother's well-being and her level of social support, as well as her level of stress as a parent and the way her children were disciplined in the family. Mother-child pairs were videotaped while completing a puzzle together and dyadic interactions were coded. Path analysis showed that paternal involvement was significantly associated with a higher level of maternal well-being and lower distress levels. In addition, higher levels of maternal distress were associated with higher levels of harsh discipline and parenting stress. Correlation analysis showed that higher perceptions of paternal involvement were associated with more positive affect displayed by the child, more positive regard for the child, and better mother-child synchrony during the dyadic interactions. Limitations include the cross-sectional design and the modest sample size, which hinder causal inferences and generalizability of the findings. These preliminary findings suggest that higher levels of paternal involvement may have an impact on markers of maternal mental health and positive mother-child interactions in families living in disadvantaged communities or

**Funding:** This research was supported by a grant from the UBS Optimus Foundation (grant number 6877, PI: JFL, https://www.ubs.com/) and partial support from the Jacobs Foundation (fellowship number 2015-1170-1, PI: LAP, https://jacobsfoundation.org/) as well as additional awards/gifts from the Open Road Alliance, ACEV and the Yale Global Health Leadership Institute. The funders had no role in study design, data collection and analysis, decision to publish, or preparation of the manuscript.

**Competing interests:** The authors have declared that no competing interests exist.

humanitarian settings. Paternal involvement should be considered when designing and implementing parenting programs in LMICs.

## Introduction

Humanitarian crises, poverty, and exposure to war, violence, and trauma place children and families at heightened risk of suffering from physical, behavioral or emotional problems [1]. Thus, there is an urgent need to identify factors that may promote the opportunities for the healthy development of children living in disadvantaged communities and refugee settings. Children who are displaced due to conflict, disasters, and other crises, are particularly vulnerable to direct trauma exposure and stressors, hazardous living conditions, unsafe environments, challenges in meeting basic survival needs (e.g., access to food, water, shelter, health care), and social isolation (e.g., [1, 2]). Recent theoretical models postulate that these adverse experiences, whether resulting from displacement or due to other circumstances, lead to toxic stress exposure—defined as frequent or sustained activation of the biological stress system without necessary buffering factors or regulatory influences—which have a negative impact on a child's neurocognitive and socioemotional development, stress response system, and the child's future behavior [3].

Therefore, appropriate family and environmental supports are necessary to mitigate the effects of toxic stress on child outcomes [4]. Specifically, improvements in parents' mental health and parenting-related stress have been linked to children's improved developmental outcomes among children who have had adverse experiences [5], particularly those families living in conflict and refugee settings [6]. Despite existing knowledge of the benefits of positive parenting for child health and development, there are numerous challenges to parenting in situations of poverty and displacement. Most of the pertinent research has been conducted with maternal caregivers and their children [7, 8]. Yet, there is evidence that paternal involvement in childrearing may serve as an important resource for families [9]. However, there is a noticeable gap in the literature with regard to the role of fathers, both in the United States and globally [10].

There are a few noticeable exceptions, including a study from Vietnam that demonstrated that fathers can be taught to interact more sensitively, responsively, and effectively with their newborn infants, and that these positive interactions promote children's motor, language, personal, and social development [11]. A population-based survey of 2,797 households in the greater Beirut area of Lebanon (including Bourj El Barajneh, one of the refugee communities in which the current study was conducted) also found that the husband's involvement in housework was associated with lower levels of maternal distress, higher levels of marital satisfaction, and overall happiness [12].

Yet, there is substantial variability in how paternal involvement is conceptualized [13]. Much of the early research embraced Lamb and colleagues' [14] influential model of the multidimensional nature of paternal involvement encompassing engagement, availability, and responsibility. Recent theoretical models (e.g., [15]) expanded upon these dimensions to further outline responsibility as indirect care and process responsibility (i.e., ensuring that family tasks are done and family norms are followed). These models are crucial as they outline the ways in which fathering may influence child development and the quality of family relationships. In fact, paternal involvement has been hypothesized as a potential modifier of the effect of parental psychopathology on child psychological adjustment via impaired parenting and maternal psychological health [13, 15].

In the present study, we operationally defined paternal involvement as engagement and parenting practices with the child and the mother, and in the community. Although theoretical models explicitly discuss father-child interactions (and paternal involvement at home more

broadly), paternal involvement in the community is a crucial factor among refugee and disadvantaged families for several reasons. Among individuals who have been forcibly displaced and exposed to war and armed conflict, the loss of social support networks may have an impact on family functioning, parental well-being, and mental health [16]. Thus, a sense of self- and community-efficacy and connectedness are crucial components to prevent the potential loss of social and community resources during human-made disasters [17].

In fact, it has been found that services connecting refugees with each other and with the broader community around them may foster community support, decrease psychological distress, and promote access to resources and employment [18, 19]. A recent study of displaced Syrians in Lebanon showed that self-settled Syrians capitalized on social networks (e.g., by evaluating community opportunities and resources available to them) to create a new livelihood system for themselves, which served as a way to cope with experiences of displacement [20]. Hence, identifying resources that were previously utilized but have been disrupted or become unavailable due to displacement (e.g., religious leaders) is an important component of working with war-affected populations [21]. Furthermore, among Arab nuclear families in Lebanon, the spousal relations and family dynamics are largely influenced by cultural and religious communities, as well as neighbors and extended communities [22]. Thus, in addition to a husband's support for his wife during activities of daily living and his involvement in childrearing, paternal involvement in the community may assist the entire family in coping with post-migration challenges and other stressors. Examples of paternal involvement in the community assessed in the present study are the extent to which the father attends community events, spends time with other fathers in the community, and enjoys spending time with other families.

Moreover, food insecurity has been linked to poor mental health (self-reported using the five-item Mental Health Inventory) in a representative sample of Palestinian refugee households in Lebanon [23]. Using the same dataset, Habib and colleagues [24] found that adults in crowded households (calculated by dividing the number of people in a home by the number of rooms, excluding the balcony, kitchen, and restrooms; above the median of 1.69 people per room was considered crowded) were more likely to report poor health outcomes—including mental health, chronic and acute illnesses, and disability—compared with those living in homes without crowding. Furthermore, in a study with over 3,000 adults from the Gaza Strip [25], perceptions of human insecurity (e.g., threats to personal safety, and fear about respondents' futures and the futures of their families) were associated with self-reported distress (e.g., individuals' fears of incapacitation and displacement, and fears of losing control over important things in their lives).

Taken together, in times of instability and prolonged periods of toxic stress imposed on the family system, fathers may provide a resource in health and adjustment for both children and mothers. Synthesized in a recent model of parenting in war, Murphy and colleagues [26] proposed several factors exerting an influence on parenting and its links to child development as a result of displacement on the level of the partner (e.g., marital tensions), household (e.g., poverty, living environment, family composition), and the community (e.g., breakdown of support networks). Despite these theoretical assumptions, there is little empirical evidence about the role of the father among refugee and other disadvantaged families in low and middle-income countries (LMICs).

## Objectives

We recently completed a randomized controlled trial (RCT) on the efficacy of the Mother-Child Education Program (MOCEP) with refugee and low-income families in Beirut, Lebanon. One of the aims of the RCT study was to identify ways to reduce risk factors for families

who were already exposed to toxic stress (as defined above). The RCT was conducted with groups of mothers and their children. However, there is emerging evidence that paternal involvement may serve as a resource to mitigate existing risk factors and to prevent (further) destabilization among families in vulnerable contexts. Thus, the objective of the present study was to examine the associations among mothers' perceptions of their husband's involvement; mothers' parenting practices (i.e., harsh discipline) and levels of parenting stress, distress, and well-being; and contextual factors (i.e., satisfaction of basic needs, social support), in families residing in disadvantaged communities and families affected by conflict, poverty, violence, and/or displacement.

Given the known association between positive mother-child interactions (e.g., better mother-child synchrony, that is, reciprocity, turn-taking, and shared affect; [27]) and child development, we also analyzed associations between paternal involvement and several indicators of mother-child interactions during a videotaped dyadic task. We hypothesized that fathers would function as a source of psychosocial support in meeting the psychological needs of both the mother and the child. Specifically, we expected to find positive associations among paternal involvement, indicators of the mother-child interaction (see Methods) and maternal well-being, and negative associations with regard to maternal distress [28]. In turn, we expected that mothers who reported higher levels of well-being and lower levels of distress would report lower levels of parenting stress and harsh discipline [29].

The loss of social support networks is considered a major displacement-related stressor that may negatively affect mental health among families affected by forced displacement [16]. Expectedly, social support has been associated with lower levels of anxiety, depression, and PTSD symptoms among resettled refugees in Australia [30] and among refugees and asylum seekers in the Netherlands [31]. Thus, we expected higher levels of social support to be associated with lower levels of maternal distress and higher levels of maternal well-being. Moreover, we hypothesized that mothers' perceptions of their personal safety, food insecurity, and household crowding would be associated with increased levels of distress and lower levels of well-being.

## Materials and methods

### Sample

The present study uses the baseline data from an RCT aimed at evaluating the impact of MOCEP on vulnerable populations in Beirut, Lebanon (ClinicalTrials.gov Identifier: NCT02402556). Participants were residents of three communities in Beirut, Lebanon: Bourj El Barajneh, Shatila, and Chiyah. Both Bourj El Barajneh and Shatila are areas of the city specifically designated for refugee families and have historically been home to members of the Palestinian refugee community. However, Shatila, in particular, has recently become home to a large number of refugees from Syria including some who are of Palestinian origin. Chiyah, by contrast, is a neighborhood in Beirut that is home to predominantly low-income Lebanese families. The directors of community and/or social service centers located within each community facilitated the recruitment process under the supervision of the MOCEP implementation coordinator. Eligibility criteria for participation in MOCEP are described in detail elsewhere (citation withheld for peer review). Mother-child dyads were invited to participate in a three-part baseline data collection procedure, in which mothers completed a series of questionnaires, children completed a series of age-appropriate activities, and dyads completed an interactive play session documented on videotape.

A total of 106 mothers participated in the RCT. Two mothers reported that there was no primary male figure within the family. Data from these mothers were excluded from the

analysis. Thus, baseline data from a total of 104 mother-child dyads were analyzed for the present study. Half of the participating children were male ($n$ = 54; 51.9%). The children were between 2.05 and 7.93 years old (mean age = 4.34 years, $SD$ = 1.24). Mothers were between 20 and 53 years old ($M$ = 32.36 years; $SD$ = 5.82). The mothers reported that they lived in their current community for an average of 20 years ($M$ = 19.64 years, $SD$ = 14.38). The range was quite broad from three months to 53 years. A total of 71 mothers (68.3%) were either born in their current community ($n$ = 21; 20.2% of the total sample), in another part of Beirut ($n$ = 27; 26% of the total sample), or in another part of Lebanon ($n$ = 23; 22.1% of the total sample); 33 mothers (31.7%) were born in another country (29, or 27.9%, were born in Syria, and 4, or 3.8%, in other Arab countries). Eighty-eight fathers (84.6% of the sample) were employed. Ninety-four mothers (90.4% of the sample) indicated that the father lived with the mother and the child. S1 Table contains additional demographic information for the sample.

## Measures

**Maternal perception of paternal involvement.**   Paternal involvement was conceptualized as the mother's perception of her husband's engagement and practices with the child, the mother, and the community. The Father Involvement Questionnaire (FIQ) has 20 items and was developed specifically for the RCT. The items were generated based on the outcomes of prior qualitative evaluations of MOCEP [32] and hypothesized markers of parental behavior that might influence family functioning. The FIQ includes nine items on the mothers' perceptions of how her husband interacts with their children (e.g., "enjoys spending time with our children" and "is not too harsh when disciplining our children."). In addition, six items pertain to the mother's perceptions of how she is treated by her husband (e.g., "usually speaks to me in a respectful manner"). The remaining five items focus on paternal involvement with other members of the community (e.g., "enjoys spending time with other families"). All items are rated on a scale from 1 (*strongly disagree*) to 4 (*strongly agree*) and were summed to compute a total score of paternal involvement (Cronbach's α = 0.82). S2 Table contains the full version of the FIQ.

**Maternal distress.**   Mothers self-reported their perceived individual distress using the Individual Distress Assessment (IDA), a 12-item rating scale used in a previous study involving a representative sample of Palestinian households in the Gaza Strip [25]. Mothers were asked to report the extent to which they are currently feeling several affects (e.g., frustrated, lonely, anxious) using a 4-point scale ranging from 1 (*not at all*) to 4 (*all the time*). A total score of individual distress was computed by averaging all items (Cronbach's α = 0.85).

**Maternal well-being.**   Mothers rated their overall well-being using the 5-item WHO-5 mental health index [33]. This brief screening assessment was developed by the World Health Organization (WHO) and is widely used in both academic and humanitarian settings. The WHO-5 asks participants to rate their subjective psychological well-being (e.g., positive mood, vitality) on a six-point scale (0 = *at no time* to 5 = *all of the time*) over the past two weeks (e.g., "I have felt cheerful and in good spirits"). The scale has been used in a study that examined the well-being of female family caretakers in two neighboring Israeli-occupied West Bank villages [34]. A total score was computed by summing all items (Cronbach's α = 0.84).

**Harsh discipline.**   The Disciplinary Style Questionnaire (DSQ; [35]) is an 18-item, parent-report measure of parental disciplinary style that assesses the frequency with which parents engage in seven forms of discipline—inductive discipline, physical punishment, manipulating privileges, harsh verbal discipline, argument, shaming, and ignoring—when dealing with children's undesirable behavior. This factor structure of the DSQ has been confirmed across nine countries, including Jordan [35]. Mothers completed the DSQ. A total score of maternal harsh discipline was computed by averaging the harsh verbal discipline (3 items) and the physical

punishment (3 items) scales (Cronbach's α = 0.61). Higher scores on this composite indicate a larger degree of harsh maternal discipline.

**Parenting stress.**   The short form of the Parenting Stress Index (PSI-SF; [36]) was used to assess mothers' perceptions of parenting stress. The PSI-SF is a 36-item questionnaire designed to measure stress within the parent-child relationship. The PSI-SF consists of three subscales: Parental Distress, Parent-Child Dysfunctional Interaction, and Difficult Child with each subscale composed of 12 items, to be rated on a five-point scale from 1 (*strongly disagree*) to 5 (*strongly agree*). The option "not sure" was inadvertently omitted (see Discussion). Scores were summed across all 36 items to compute a total score of parenting stress (Cronbach's α = 0.85). Higher scores indicate greater levels of stress.

**Basic needs.**   Three types of basic human needs were assessed. Mothers' perceptions of their personal safety and security, as well as that of their families, were assessed using the 10-item Human Security Assessment (HSA; [25]). The questions take into account participants' culture and value systems when considering their standards of and concerns about the safety of their surroundings. The food security of participating families was assessed with eight items using the Arab Family Food Security scale (AFFS; [37]) plus one supplemental question from the evaluation of the Healthy Kitchens, Healthy Children project, a study in Beirut, Lebanon in which the participants were members of the Palestinian refugee community [23]. The food security questionnaire was developed specifically for Arab communities in the Middle East and measures degrees of family-level food security in the previous year. Finally, crowding was measured by dividing the total number of people living in the household by the total number of rooms in their home environment. Scores on the AFFS showed small positive correlations with the HSA ($r = 0.25$, $p < 0.05$) and the household crowding index ($r = 0.29$, $p < 0.01$). There was a small positive correlation between the HSA and the household crowding index ($r = 0.27$, $p < 0.05$) as well. A principal factor analysis using the three basic needs variables showed 27.57% of variance explained by a single factor. The factors loadings were 0.53, 0.50, and 0.55 for AFFS, HSA, and household crowding, respectively. The factor score was saved for further analysis using a least squares regression approach. Higher factor scores indicate higher levels of socioeconomic need.

**Social support.**   The Duke Social Support Index, Short Form (DSSI-SF; [38]), assesses social support in two domains: social interactions and subjective support. Recently, the abbreviated, 11-item version was used in the Healthy Kitchens, Healthy Children Project [39]. The first four items assess social interactions. One item asks the mother to indicate the number of persons in her area other than her family that she feels she can depend on or feels very close to (1 = *none* to 3 = *more than two people*). Three items require the mother to rate (a) how many times during the past week she spent time with someone who does not live with her, (b) how many times she talked to someone on the telephone, and (c) how often she attended meetings of clubs, religious meetings, or other groups. The original answer scale for these three items ranged from 0 (*none*) to 7 (*7+ times*). As per the developers' guidelines, the answers for item (a) above were recoded to 1 = *none*, 2 = *one time or two times*, and 3 = *three or more times*. The answers for items (b) and (c) above were recoded to 1 = *none or one time*, 2 = *between two and five times*, and 3 = *six or more times*. The remaining seven items assess subjective support and were rated on a scale from 1 (*hardly ever*) to 3 (*most of the time*). Father support was not directly assessed using this scale; yet, the subjective support items asked the mother to rate the frequency of support received from her family more broadly, which includes her husband. A total score of social support was computed by summing the scores from the 11 items (Cronbach's α = 0.72).

**Dyadic (mother and child) interactions.**   Mothers and children were provided with a wooden puzzle and were asked to play together for 10 minutes. Prior to starting, mothers and

children were told (in Arabic) that there was no "right" or "wrong" way to use the puzzle and were asked to play as they would at home. Interactions were videotaped by research assistants, who remained in the room during the interaction. To reduce bias, mothers were made comfortable and instructed by the researchers to just act "naturally." Also, mothers who did not want to show their face to the video camera did not have to do so. The videotaped interactions were translated and subtitled for behavioral coding by a native Arabic speaker with relevant credentials (M.D., M.Sc.). Mothers were coded for maternal sensitivity, intrusiveness (for data analysis, this scale was reverse-coded to reflect unobtrusive behavior of the mother), positive regard, and disengagement during the interaction, following criteria established by Bernard, Meade, and Dozier [40]. Children were coded in terms of positive affect, negative affect, and persistence during the task, following criteria established by Matas, Arend, and Sroufe [41]. Dyadic synchrony was operationally defined as dyadically regulated affective exchanges and connectedness, according to criteria as in Harrist et al. [42]. High dyadic scores were provided for high positive dyadic synchrony. Low scores were provided for disconnected and asynchronous interactions as well as for mutual displays of negative affect between parent and child. All videos were double coded. Twenty-five percent of the videos were also triple coded by the lead trainer to ensure consistency in coders' ratings. With regard to inter-rater reliability, correlations between coders ranged from $r = 0.85$ to $0.92$. A total of 92 families had baseline data from both the FIQ and the dyadic interactions.

## Procedure

Data collectors were female researchers from Lebanon experienced in direct assessments and survey methods, with a background in psychology. Assessors were trained in Beirut by two Yale researchers, and the protocol was piloted prior to rolling out the trial. Data used for the present study were collected over four weeks in each of the three sites over the course of three months. An average of three dyads was interviewed per day, followed by a dyadic interaction task. Each questionnaire took about 45 to 90 minutes to complete, followed by 15 minutes for the recording of the dyadic interaction. Recruitment to the study was conducted in close partnership with the community centers, and included sending letters, holding community gatherings and town hall meetings, and following up as needed via telephone calls. The Principal Investigator in Lebanon (GI) obtained written consent from center directors to ensure proper understanding of the study's purpose, procedures, rights and benefits to all participants. Interested mothers who were eligible to participate were invited to an introductory group meeting where they were scheduled for their initial assessment (baseline), at which time written consent and permission for the children to participate in the study were obtained. Upon consultation with local partners, monetary compensation for each data collection time point was deemed the appropriate incentive. This trial was conducted according to the guidelines of the Human Subjects Committees at Yale University and the Arts, Sciences, and Technology University in Lebanon, who approved all recruitment, assessment, data management, and participant protection procedures. No adverse effects were reported during the study.

## Data analysis

Data were analyzed in two steps. First, we analyzed descriptive statistics and zero-order correlations for the main study variables. Second, we fit a path model to the data to test for indirect effects of paternal involvement on maternal parenting stress, discipline, and dyadic interaction via maternal individual distress and well-being, respectively. Basic needs and social support were included as control variables to partially address the problem of potential mediator-outcome confounding. Direct and indirect effects were estimated in *Mplus* 7.4 [43] using full

information maximum likelihood (FIML). FIML was also used to handle missing data. Non-symmetric bootstrap (50,000 bootstrap draws) confidence intervals for indirect effects were computed because of the non-normal distribution of indirect effects due to the relatively small sample size. Individual distress and maternal well-being were tested as parallel mediators. The overall goodness-of-fit of the path model was evaluated based on the following fit indices: the comparative fit index (CFI), the standardized root mean square residual (SRMR), and the root-mean-square error of approximation (RMSEA). Estimates of the CFI close to 0.95, SRMR close to .08, and RMSEA close to .06 indicate adequate model fit [44].

## Results

### Maternal well-being and parenting, and their associations with paternal involvement

Data for individual variables were missing for 0 to 9 (8.7%) participants. Little's MCAR test indicates that data were missing completely at random, $\chi^2$ (121) = 113.50, $p$ = .67. Table 1 presents descriptive statistics and correlations. The mother's perception of her husband's involvement showed small negative, statistically non-significant correlations with maternal parenting stress (PSI) and harsh discipline (DSQ). Moreover, more involvement of fathers showed a small association with lower levels of socioeconomic (basic) needs. Moreover, higher levels of socioeconomic (basic) needs showed medium correlations with higher levels of maternal distress and lower levels of well-being, thus confirming our hypothesis. Higher maternal perceptions of social support showed small correlations with lower levels of distress and higher levels of well-being, which supports our hypothesis about the role of social support.

With regard to the proposed mediating variables, higher maternal perception of paternal involvement showed a medium correlation with lower levels of maternal distress (IDA) and higher levels of well-being (WHO-5). Higher levels of well-being and lower levels of distress, in turn, showed small (well-being) to medium (distress) associations with maternal harsh discipline and parenting stress, respectively, which supports our hypothesis. Maternal harsh discipline showed a medium and positive relation with perceived parenting stress.

Fig 1 shows the path model testing the indirect effect of the father's involvement, as reported by his wife, on the mothers' harsh discipline, her level of parenting stress, and the

**Table 1. Descriptive statistics and correlations for maternal self-reports.**

| | Baseline ratings | 1. | 2. | 3. | 4. | 5. | 6. | 7. |
|---|---|---|---|---|---|---|---|---|
| 1. | Maternal perception of paternal involvement | | -0.24* | 0.09 | -0.40*** | 0.40*** | -0.20 | -0.18 |
| 2. | Basic needs | | | -0.42*** | 0.49*** | -0.20** | 0.25* | 0.31** |
| 3. | Social support | | | | -0.28** | 0.27** | -0.12 | -0.32** |
| 4. | Maternal distress | | | | | -0.51*** | 0.41*** | 0.44*** |
| 5. | Maternal well-being | | | | | | -0.22* | -0.29** |
| 6. | Maternal harsh discipline | | | | | | | 0.54*** |
| 7. | Maternal parenting stress | | | | | | | |
| | Mean | 52.69 | 0.00 | 23.21 | 2.35 | 8.49 | 7.40 | 109.11 |
| | SD | 5.29 | 0.74 | 4.18 | 0.58 | 6.33 | 2.47 | 15.01 |
| | Missing (n, %) | 2, 1.90 | 4, 3.90 | 0 | 9, 8.70 | 0 | 5, 4.80 | 0 |

N = 104. Higher scores on the basic needs composite indicate higher socioeconomic needs.

*p < .05.

**p < .01.

***p < .001.

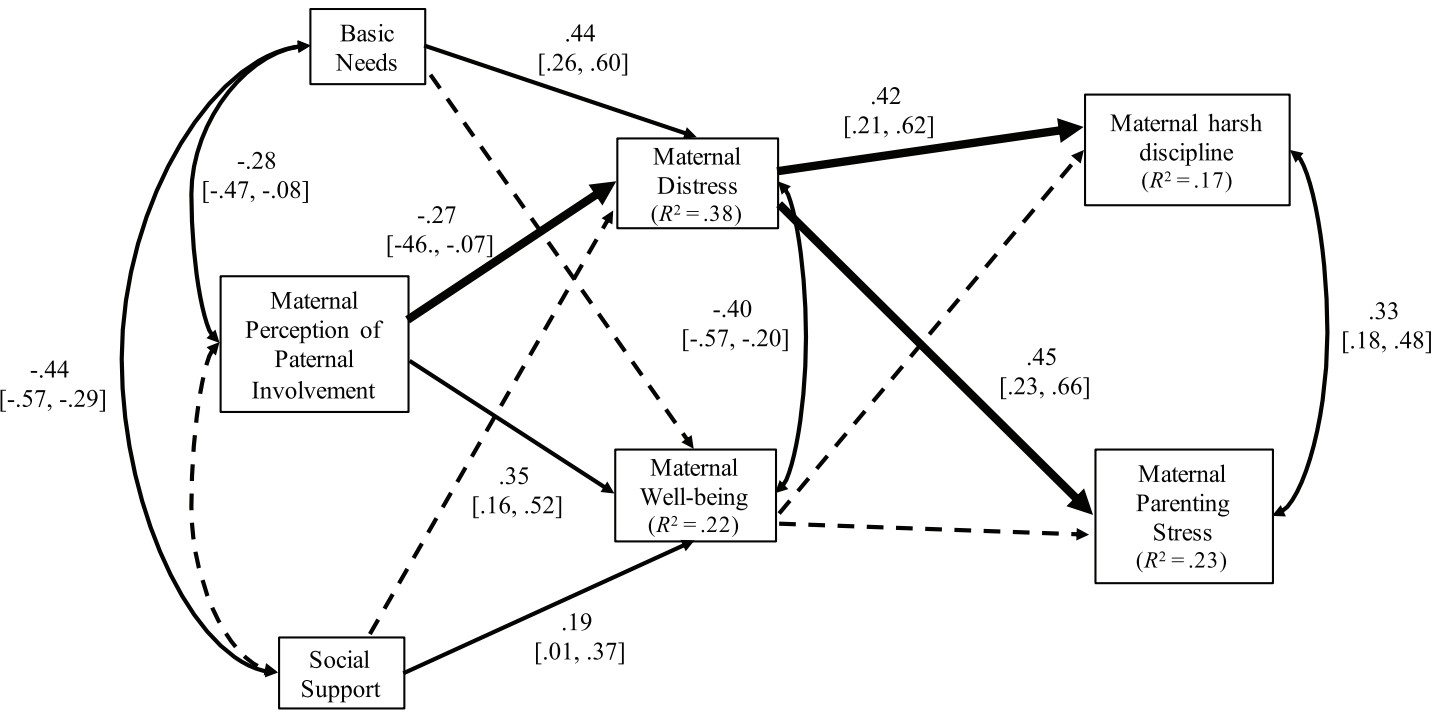

**Fig 1. Path model testing the indirect effect of paternal involvement, as reported by the mothers, on her harsh discipline (a composite score of harsh verbal discipline and physical punishment) and her level of parenting stress.** Standardized estimates are reported with 95% confidence intervals in parentheses. Dotted paths indicate statistically non-significant paths. The thick lines indicate the indirect effects from paternal involvement to maternal harsh discipline and the mother's perceived parenting stress via maternal distress that were significant at the 0.05 level. Higher scores on the basic needs variable indicate higher socioeconomic need.

dyadic interaction composite. This model yielded an acceptable fit to the data, $\chi^2$ ($df = 6$) = 5.81, $p = .45$, RMSEA = 0.000, CFI = 1.00, SRMR = 0.033. Standardized estimates showed that higher levels of paternal involvement were associated with less maternal distress and more well-being, controlling for basic needs and social support. Higher levels of socioeconomic needs, as measured by the extent to which families "basic needs" were not met, were related to higher levels of maternal distress but unrelated to maternal well-being. Even after taking into account the strong link between perceived paternal involvement and maternal well-being, the associations between social support and mothers' well-being was small but statistically significant. Maternal perceptions of social support, however, were not significantly related to ratings of distress.

The indirect effect estimates showed a small, statistically significant overall indirect effect of paternal involvement on parenting stress via maternal distress and well-being ($B = -0.39$, $SE = 0.15$; $\beta = -0.14$, $p < .01$; 95%-CI = -0.24 to -0.04). The decomposition of the total indirect effect into specific indirect effects showed that the total indirect effect was mainly attributable to maternal distress ($B = -0.34$, $SE = 0.15$; $\beta = -0.12$, $p = .02$; 95%-CI = -0.23 to -0.03) but not to well-being ($B = -0.05$, $SE = 0.11$; $\beta = -0.02$, $p = .63$; 95%-CI = -0.10 to 0.05). The overall indirect effect of paternal involvement on maternal harsh discipline was small and negative ($B = -0.05$, $SE = 0.02$; $\beta = -0.11$, $p = .01$; 95%-CI = -0.20 to -0.02). The decomposition of this indirect effect showed a stronger indirect effect via maternal distress ($B = -0.05$, $SE = 0.02$; $\beta = -0.11$, $p = .02$; 95%-CI = -0.22 to -0.03) compared to well-being ($B = 0.00$, $SE = 0.02$; $\beta = 0.01$, $p = .90$; 95%-CI = -0.07 to 0.08). A series of tests of the differences in indirect effects showed that the indirect effect of paternal involvement on parenting stress via maternal distress was stronger compared

to the indirect effect of paternal involvement on harsh discipline via maternal distress, $B = 0.29$, $SE = .131$, $p = .027$). The differences in other indirect effects of paternal involvement on harsh discipline and parenting stress via maternal distress and well-being were small and statistically non-significant.

### Mother-child interactions, and their associations with paternal involvement

Among the eight indicators of mother-child interactions obtained from the dyadic interaction task (Table 2), paternal involvement showed small and positive associations with positive regard ($r = 0.23$, $p < .05$), synchrony ($r = 0.29$, $p < .01$), and the child's positive affect ($r = 0.28$, $p < .01$). Correlations between the indicators of mother-child interaction and mothers' self-reported well-being and parenting were overall small and statistically non-significant (data not shown). The only statistically significant correlation was between well-being (WHO-5) and the child's positive affect ($r = .23$, $p < .05$), indicating that children of mothers who reported higher levels of general well-being displayed more positive affect during the dyadic interaction task. These findings partially support our expectations to find associations between paternal involvement, indicators of the mother-child interaction, and maternal well-being and distress.

### Discussion

The findings of the present study indicate that fathers can be an important source of support for mothers in resource-constrained communities affected by conflict, displacement, and marginalization. In this study, the mothers' perceptions of greater paternal involvement were significantly associated with a better sense of subjective psychological well-being and lower levels of personal distress. In support of our hypothesis, we found that paternal involvement was positively associated with maternal well-being and negatively associated with maternal distress. In addition, we found partial support for the hypothesis that mothers who reported higher levels of distress (but not well-being) reported lower levels of parenting stress and harsh discipline. Thus, although there were no direct relations between paternal involvement and maternal harsh discipline and parenting stress, there were indirect relations via maternal perceptions of distress. The negative association between paternal involvement and maternal psychological distress (e.g., maternal depressive symptoms), has been well-documented in research on the influence of the relations with nonresident fathers on the well-being of mothers (specifically among non-employed mothers; [45]), perceptions of support from mothers' partners who are

**Table 2. Descriptive statistics and correlations for indicators of mother-child interactions.**

| Variable | M | SD | Correlation with paternal involvement |
|---|---|---|---|
| Sensitivity | 2.94 | 1.01 | 0.20 |
| Unobtrusive behavior | 3.79 | 1.11 | 0.17 |
| Disengagement | 1.09 | 0.48 | -0.18 |
| Positive Regard | 2.95 | 1.03 | 0.23* |
| Synchrony | 2.92 | 1.09 | 0.29** |
| Child Negative Affect | 1.60 | 0.80 | -0.14 |
| Child Non-Compliance | 1.21 | 0.62 | -0.09 |
| Child Positive Affect | 2.86 | 0.93 | 0.28** |

$n = 92$ (data from 12 mother-child dyads were missing). The range of possible scores on all variables is 1–5.

*$p < .05$.

**$p < .01$.

not formally identified as stepfathers (e.g., among Mexican-origin mothers with 3-year-old children; [46]), and the contributions of the quality of the marital relationship (e.g., the extent of agreement between partners on a range of family-related issues) to maternal well-being among mothers of children with developmental disabilities [47]. Similar to the findings of the present study, maternal distress and parenting stress are often interrelated [48–50], whereas evidence supporting a direct link of paternal involvement/support and perceptions of parenting/parental stress is less robust [45, 46].

We drew from the existing literature in relatively comparable populations, and our data indicated that a composite assessing food and human insecurity and household crowding index, was a valid proxy to assess a mother's ability to meet her/her family's basic needs. As expected, our findings indicated that an inability to meet the family's basic needs was associated with distress among mothers. These findings are consistent with a substantive body of literature linking socioeconomic need and maternal distress during infancy [51] and through early childhood development [52]. Importantly, maternal distress was associated with parenting stress and harsh discipline in our sample, suggesting that structural challenges are related to parental functioning and behaviors. These findings implicate economic and food security supports as key strategies to accompany targeted social interventions.

It is clear that a child's well-being is fundamentally tied to the well-being of their primary caregivers, typically their mothers, and that the mothers' well-being rests fundamentally on the supports (e.g., psychological, emotional support) that they receive from people in their everyday lives as well as the quality of the relationship (e.g., the level of conflict, cooperation) between the primary caregivers (e.g., [53]). The findings of our study support this notion. As evidenced by the bivariate correlations, higher levels of perceived social support were associated with higher levels of subjective maternal well-being and lower ratings of maternal distress. Thus, although our focus was on father-mother relations, we fully acknowledge that supportive relations manifest among wider networks. This reality is also discussed at length in the recent comprehensive report from the National Academies of Science, Engineering, and Medicine, entitled "Vibrant and Healthy Kids" [54]. For instance, the authors of the report recommend that "special attention should be given to the recruitment, retention, and support of men and fathers parenting young children from underserved populations" (p. 9). We would like to underscore this recommendation, particularly with regard to fathers from refugee and marginalized communities.

Our findings resonate well with theoretical models emphasizing the importance of the partner in the relationship among parental stress, psychological well-being, and parenting behaviors [14, 15, 26]. Interestingly, the mother's perception of paternal involvement was not directly associated with ratings of how frequently she used harsh discipline when dealing with her child's undesirable behavior. One possible explanation for this finding is the idea that paternal involvement is affected by maternal characteristics (i.e., "maternal gatekeeping") such as the reluctance to relinquish family responsibility by setting rigid standards, a desire to validate a maternal identity, and differentiated conceptions of family roles [55].

In their seminal research on maternal gatekeeping, Allen and Hawkins [55] conceptualized maternal gatekeeping as three interrelated perceptions and behaviors: (a) mothers' reluctance to relinquish responsibility over family matters by setting rigid standards; (b) external validation of a mothering identity; and (c) differentiated conceptions of family roles. More recently, Puhlman and Pasley [56, 57] proposed a three-dimensional model of gatekeeping composed of encouragement (i.e., facilitative dimension; gate open), discouragement (i.e., inhibitory dimension; gate closed) and control. Encouragement is characterized by positive feedback (e.g., expression of positive regard through compliments) and welcoming behaviors of the mothers, which support father involvement with children. Discouragement, on the other

hand, entails maternal expression of negativity toward fathers through behaviors such as frequent criticism and avoidance of co-parenting interactions.

Collectively, research points to standard and responsibility, maternal identity confirmation, differentiated family roles, encouragement, discouragement, and control as crucial dimensions of maternal gatekeeping. Additional research examined antecedents of maternal gatekeeping and father involvement and highlighted the differential roles of fathering identity (i.e., the salience of the role as a father) and maternal influences (e.g., psychological correlates such as personality, negative emotionality, self-esteem) as well as life transitions and social context (e.g., the socially scripted role of the mother), among other factors [58–60]. In line with this previous research, mothers in our study who reported high levels of well-being and low levels of distress may be more likely to encourage fathers to take responsibility for family matters. Conversely, low levels of well-being and high levels of distress may facilitate inhibitory gatekeeping behaviors such as discouragement as a way to gain appreciation from one's social context(s). Together, these assumptions point to complex, dynamic and reciprocal factors that, to our knowledge, have not been examined in relation to experiences of displacement and migration. Drawing from recent research on the associations between supportive relationships and caregiver well-being [53], future research may seek to examine the role of caregiver well-being in curtailing maternal gatekeeping and promoting cooperation among caregivers in general [61], and in refugee and marginalized communities in particular.

With regard to the observed mother-child interactions, we expected to find positive associations among paternal involvement, indicators of the mother-child interaction and maternal well-being, and negative associations with regard to maternal distress. Supporting this hypothesis, we found that higher levels of mother's perception of paternal involvement were significantly related to more positive affect displayed by the child and better synchrony between the mother and her child during the interaction. As expected, higher levels of paternal involvement were related to lower levels of maternal distress. However, maternal distress was not directly associated with behaviors and affects during the interaction task. Intriguingly, we also found that the child's positive affect during the interaction was positively related to mothers' well-being (WHO-5). Together, these findings echo the idea that better mother-child synchrony is associated with a healthy mother [27] and that fathers may provide a source of support that affects positive interactions between the mother and her child. The latter conclusion was supported, in part, by the (small) correlations between paternal involvement and maternal and child behavior during the interaction task. However, maternal well-being was not significantly associated with the indicators of observed mother-child interactions. Thus, future studies should examine the characteristics and factors that may mediate the association between paternal involvement and mother-child interactions.

The maternal ratings of perceived social support were only slightly and not statistically significantly related to perceptions of paternal involvement ($r = 0.09$). Findings from the path model showed that paternal involvement was related to both maternal distress and maternal well-being when controlling for basic needs and perceptions of social support. Whereas, maternal ratings of perceived social support were only related to maternal well-being but not to maternal distress when controlling for basic needs and paternal involvement. These findings support the notion that both constructs may function as distinct sources to promote maternal well-being. Together, these findings support the notion that maternal distress may mediate the association between maternal perceptions of paternal involvement and both maternal parenting stress and maternal harsh discipline.

## The cultural context for paternal involvement and family functioning in Lebanon

Family is a fundamental social institution and central to the lives of people in Middle Eastern communities [62–64]. However, due to differences in factors such as the family's religious affiliation, place of residence, and national affiliation, it is difficult to pinpoint one model that typifies all Arab families [22], and that, therefore, could allow for the determination of normative relational patterns among refugee and disadvantaged families in Lebanon. Yet, it has been argued that the Lebanese family has evolved from an extended family structure with the male elder as the head of the family (i.e., patriarchal family) to a nuclear family structure (i.e., consisting of the parents and their dependent children) [63, 65]. The nuclear family in Lebanon is horizontal in its interfamilial relationships, and interdependence (e.g., in the form of solidarity, cohesion, and the marital union) is considered a desirable attribute among its members [64, 65]. Not unique to families in the Middle East, paternal involvement in childrearing, in the mother-child relationship, and in the community may be interpreted by the mother, as a demonstration of sympathy and of her commitment to and bond with the family unit, which could positively affect her well-being. In our data, this conclusion is, in part, supported by the finding that paternal involvement was positively related to maternal well-being. Conversely, low levels of paternal involvement conflicts with the cultural expectation of the husband to assume responsibility for the protection of the family and to provide economic welfare (e.g., by being more involved in the community to secure employment), thus having a negative impact on the stability and well-being of the family unit [64, 66]. The finding that lower levels of perceived paternal involvement were associated with higher levels of maternal distress substantiates this notion.

Although we did not explore factors contributing to paternal involvement, it has been shown that times of economic strains and hardship increase the additional burden for Arab women to participate in the formal labor market [67] and challenge the traditional role expectation of the husband as provider for the family. Moreover, economic strains may have an impact on the husband's standing in the community and the family, including loss of status and authority [63, 68]. Recent research from high-income countries indicates that there has been a convergence of men's and women's roles in the household, particularly in terms of family work, including childcare [69]. Thus, the increased burden experienced by mothers with the least supportive spouses may also relate to higher distress and poorer subjective psychological well-being. Further research is needed to investigate sources of differences in paternal involvement among families living in poverty and humanitarian settings. In this regard, we need to be careful not to generalize the findings across culture as the literature on cross-cultural co-parenting is scarce, and recent reviews have highlighted the need for additional research globally on paternal engagement in parenting interventions [70].

Conflict and war are disruptive and harmful for families [26]. Yet, growing up affected by conflict may elicit collaboration among women in refugee camps [71]. For example, after the Lebanese civil war, Joseph [66] observed the paradoxical situation that Lebanese women and youth were in some ways empowered during the war by this greater need for family in the absence of public institutions that could protect a person's security and, in turn, their well-being. Thus, in these contexts, increased paternal involvement may compensate for the harmful effects of the loss of social support networks [16], and may promote access to resources that could decrease psychological distress. In fact, the findings from the present study support that mothers' perception of paternal involvement were associated with improved self-reported well-being and mental health, as well as observations of mother-child interactions among families living in poverty or humanitarian settings. Future studies should examine the extent and

nature of additional resources available to the family due to increased paternal involvement, and whether these resources positively impact maternal well-being.

## Limitations

The cross-sectional design of this study along with the modest sample size hinder causal inferences and generalizability of the findings. Moreover, a relatively large number of tests were conducted based on this small sample. Thus, the reported levels of statistical significance should be reported with caution. Future studies need to include the fathers' personal assessment of their involvement with their children, wives, and communities before and after any interventions. Ideally, this would include collecting both quantitative and qualitative data. This is especially important, as some of the fathers' behaviors may be unknown to the mothers (e.g., the extent to which a father spends time with other fathers in the community, or the extent to which he enjoys spending time with other families). Thus, it is crucial for future studies to include the father's own ratings of their involvement as well as the ratings of their children, as they may emphasize dimensions from a different perspective when asked to report on their fathers' involvement [72]. It will be especially interesting to see how well-aligned the mother's perceptions of her husband's role in the family are with the father's own perceptions, as well as the perceptions of the child. However, it is important to consider aspects related to maternal gatekeeping when attempting to assess the fathers' perceived involvement, as studies have found that mothers' beliefs about the role of fathers regulate the relationship between fathers' perceived roles and actual levels of involvement [59].

Furthermore, the presence of a researcher during the mother-child interaction may hinder naturalistic behaviors of the mother and the child, which might explain low ratings of maternal disengagement and intrusiveness. Moreover, we cannot rule out the possibility of bias due to (a) administering questionnaires prior to observing the mother-child interactions, and (b) translating the videotaped interactions from Arabic into English. Future studies should aim to validate the coding protocols and procedures with Arabic-speaking populations in similar contexts. There is also a clear need to assess the psychometric properties of the FIQ and other relevant assessment tools with larger samples. The sample size was not sufficient to conduct a factor analysis of the FIQ. Consequently, it was not possible to sufficiently disentangle the differential effects of paternal involvement with their wives, their children, and the community on maternal well-being and parenting. Moreover, the "not sure" option was inadvertently deleted from the Arabic version of the PSI-SF, so that the instrument had just a four-point scale. However, the reliability of the measure was acceptable.

More research is also needed on the validation of the measures among disadvantaged and refugee families in Lebanon. For instance, there may be distinct profiles of psychological distress that may be more susceptible to the qualities of the mother-father relationship than others. Finally, future research should expand on the model presented here by examining the impact of paternal involvement on child development outcomes via maternal well-being and parenting behaviors.

## Future directions and conclusion

Some of the most promising findings across studies examining the effects of parenting interventions in LMICs are reduced use of dysfunctional parenting practices, decreased parental stress, increased maternal well-being, and increased awareness of positive parenting strategies and appropriate disciplinary techniques [73, 74]. Here, we identified paternal involvement as an important (indirect) correlate of mothers' perceived parenting stress that should be considered when designing and implementing early childhood parenting programs. There is

evidence that paternal involvement can be promoted. For instance, Pruett, Pruett, Cowan, and Cowan [61] examined the effectiveness of the Supporting Father Involvement (SFI) intervention—a 16-week couple-focused paternal involvement intervention for low-income families—in an RCT in five California counties. The authors found that parents who participated in a couples group reported that fathers spent more time with the child. Couples also reported reductions in parenting stress, and less use of violent problem solving in the family. Additional culturally sensitive research is needed to explore the feasibility for implementing such co-parenting and couple relationship programs with mothers and fathers in humanitarian settings.

In our RCT with families in Beirut, Lebanon, we found that mothers' participation in MOCEP did not improve their perceptions of their husbands' involvement as measured with the FIQ [75]. Fathers are generally understudied in these programs and should be more purposefully included in the design and delivery, as well as in the process and impact evaluation of such programs [70]. However, there may be important personal, familial, and societal barriers to address in order for men to become more involved in child rearing in general, and in (co-) parenting programs in particular [8].

Given these complex—and, oftentimes, intertwined—factors, how can we support fathers in navigating these issues so that they can be more involved with their families? We interviewed 13 fathers after their wives and children had participated in the MOCEP. One basic obstacle that some fathers pointed out was their lack of availability due to work schedules. Non-traditional means of recruitment and outreach may be necessary to reach fathers who are least likely to be involved. Anecdotal comments from fathers in refugee settings revealed various unique concerns and worries that may prevent fathers from being maximally involved with their family. Common stressors include a feeling of powerlessness due to underemployment and lack of ability to provide for one's family. Perceived isolation and discrimination from the larger community may also highly affect the fathers' well-being. Together, these extreme stressors may affect fathers' well-being more generally, which can compromise their levels of involvement. These anecdotal observations may reveal key issues that need to be targeted in any program that attempts to enhance paternal involvement in disadvantaged communities and humanitarian settings. These are critical issues from the perceptive of the fathers we interviewed. However, they are neither solely nor easily solved at the level of family-based intervention. Among many factors, cultural, gender, and migration dynamics, as well as community norms and national policies need to be considered to devise strong policy and/or program recommendations, which are outside the scope of this work. The incremental contribution of this study is to substantiate the role of paternal engagement and support, with further research needed to develop effective programs and strategies that may reduce the major concerns voiced by fathers.

Although the present study did not aim to ascertain predictors of paternal involvement, the (quantitative) findings point to the key role of paternal involvement for maternal outcomes. Thus, the perceptions and feelings described by the fathers in the interviews point to potential factors contributing to paternal involvement that should be the focus of future studies.

Moreover, we know little about fathers' desires and perceptions related to parenting, or about their motivations to engage with the community and the family in disadvantaged communities and humanitarian settings. For example, some of the fathers have lived in these communities for their entire lives and have ample knowledge of approaches that have failed and succeeded in the past. More monitoring of paternal involvement with their families and their communities is required, as are independent impact evaluations of programs that seek to improve opportunities for involved fathering. Given this anecdotal evidence, there is a need to determine the extent to which the families' involvement in the MOCEP impacted the associations found in the present study.

In conclusion, we would like to offer two calls to action. First, research, monitoring, and evaluation in humanitarian settings should pay greater attention to co-developing robust, context-specific, and time-specific tools that capture paternal involvement and the relation with maternal well-being and parenting together with families living in these areas. Second, after such tools have been developed and tested, there should be targeted efforts to translate the knowledge gained into evidence-based capacity building and actionable recommendations for practitioners. First and foremost, such an initiative would contribute to the discourse about the need for paternal involvement interventions in humanitarian settings. The translation of such knowledge should occur in collaboration with the families and communities themselves.

## Supporting information

**S1 Table. Sample demographic characteristics.**
(PDF)

**S2 Table. Father Involvement Questionnaire (FIQ).**
(PDF)

## Acknowledgments

We are grateful for Kyle Pruett's thoughtful questions and comments on earlier drafts of this manuscript, to Nancy Shemrah Fallon for her support in data management and leading ethics review procedures for the project, as well as to Redha Qabazard who conducted the father interviews. We are indebted to the community leaders for enabling the implementation and evaluation of the program, as well as to the families who participated in the research. We are also thankful for the support of the Mother-Child Education Foundation (AÇEV) during the implementation and evaluation of this project.

## Author Contributions

**Conceptualization:** Sascha Hein, Johanna Bick, Kaveh Khoshnood, Rima Salah, Pia R. Britto, James F. Leckman, Liliana Angelica Ponguta.

**Data curation:** Sascha Hein, Anna L. Zonderman, Sarah Householder, Liliya Katsovich, Christina Moore.

**Formal analysis:** Sascha Hein.

**Funding acquisition:** Ghassan Issa, James F. Leckman, Liliana Angelica Ponguta.

**Investigation:** Sascha Hein, Ghassan Issa, Lara Aoude, Cosette Maalouf, Abir Awar, Sawsan Nourallah, Kaveh Khoshnood, Rima Salah, James F. Leckman, Liliana Angelica Ponguta.

**Methodology:** James F. Leckman, Liliana Angelica Ponguta.

**Project administration:** Ghassan Issa, Lara Aoude, Cosette Maalouf, Abir Awar, Sawsan Nourallah, James F. Leckman, Liliana Angelica Ponguta.

**Resources:** Ghassan Issa, James F. Leckman, Liliana Angelica Ponguta.

**Software:** Sascha Hein.

**Supervision:** Ghassan Issa, Rima Salah, Pia R. Britto, James F. Leckman, Liliana Angelica Ponguta.

**Visualization:** Sascha Hein.

**Writing – original draft:** Sascha Hein, Johanna Bick, Anna L. Zonderman, Sarah House-holder, Liliya Katsovich, Kaveh Khoshnood, James F. Leckman, Liliana Angelica Ponguta.

**Writing – review & editing:** Sascha Hein, Johanna Bick, Anna L. Zonderman, Sarah House-holder, Liliya Katsovich, Kaveh Khoshnood, James F. Leckman, Liliana Angelica Ponguta.

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
