## [Decision Letter · Decision Letter 0]

4 Nov 2019

PONE-D-19-24487

Maternal perceptions of father involvement among refugee and disadvantaged families in Beirut, Lebanon

PLOS ONE

Dear Dr. Hein,

Thank you for submitting your manuscript to PLOS ONE. After careful consideration, we feel that it has merit but does not fully meet PLOS ONE’s publication criteria as it currently stands. Therefore, we invite you to submit a revised version of the manuscript that addresses the points raised during the review process.

We would appreciate receiving your revised manuscript by Dec 19 2019 11:59PM. To enhance the reproducibility of your results, we recommend that if applicable you deposit your laboratory protocols in protocols.io, where a protocol can be assigned its own identifier (DOI) such that it can be cited independently in the future. For instructions see: http://journals.plos.org/plosone/s/submission-guidelines#loc-laboratory-protocols

We look forward to receiving your revised manuscript.

Kind regards,

Florian Fischer

Academic Editor

PLOS ONE

Journal Requirements:

2. Please provide additional details regarding participant consent. In the ethics statement in the Methods and online submission information, please ensure that you have specified whether consent was written or verbal/oral. If consent was verbal/oral, please specify: 1) whether the ethics committee approved the verbal/oral consent procedure, 2) why written consent could not be obtained, and 3) how verbal/oral consent was recorded. If your study included minors, please state whether you obtained consent from parents or guardians in these cases.

3. We noted in your submission details that a portion of your manuscript may have been presented or published elsewhere. [In this paper, we analyzed baseline data from 104 mother-child dyads who participated in a randomized controlled trial aimed at evaluating the impact of the Mother-Child Education Program on vulnerable populations in Beirut, Lebanon. We used baseline and endpoint data from the same trial to write a paper on the outcome evaluation of a randomized controlled trial concerning the effects of the Mother-Child Education Program on parenting stress and disciplinary practices among refugee and marginalized communities in Lebanon (considered for publication and currently under review in Journal of the American Academy of Child & Adolescent Psychiatry). Moreover, we conducted a process evaluation of the intervention (accepted for publication in the journal New Directions for Child and Adolescent Development). Both manuscripts are attached to this submission as supplemental material.]

* Please clarify whether this  publication was peer-reviewed and formally published. If this work was previously peer-reviewed and published, in the cover letter please provide the reason that this work does not constitute dual publication and should be included in the current manuscript.

Reviewers' comments:

Reviewer's Responses to Questions

**Comments to the Author**

1. Is the manuscript technically sound, and do the data support the conclusions?

Reviewer #1: Partly

Reviewer #2: Partly

2. Has the statistical analysis been performed appropriately and rigorously? 

Reviewer #1: Yes

Reviewer #2: No

3. Have the authors made all data underlying the findings in their manuscript fully available?

Reviewer #1: Yes

Reviewer #2: Yes

4. Is the manuscript presented in an intelligible fashion and written in standard English?

Reviewer #1: Yes

Reviewer #2: Yes

5. Review Comments to the Author

Reviewer #1: This manuscript covers an important and interesting topic in an often-overlooked population – relations between paternal involvement, contextual factors, and maternal parenting/outcomes in a sample of refugee and socioeconomically disadvantaged families. Please see my comments below for suggestions for improving the manuscript for publication.

1. Conclusions from the data

1. The authors appear to have used face-valid and relatively reliable measures (given the limitations of researching this population) and conducted appropriate analyses. However, their interpretation of their findings in the Discussion would benefit from more explicit integration of their findings in the context of research and practice.

2. Although there were no direct relations between paternal involvement and maternal harsh discipline, there were indirect relations. The Discussion would be strengthened by discussing these pathways and their implications for research and practice.

3. How do you propose that maternal gatekeeping relates to paternal involvement and subsequent maternal harsh discipline?

4. The statement that “fathers may provide a source of support that affects positive interactions between the mother and her child” were not supported by the data because indirect relations between paternal involvement and maternal well-being, via mother-child synchrony, were not assessed.

5. The Discussion would be strengthened by the inclusion of your findings on relations between basic needs and maternal outcomes. What are the implications of your findings?

6. Please elaborate about the role of perceived social support (page 20, last paragraph). Is your argument that both perceived social support and paternal involvement were separately related to maternal outcomes, but not strongly related to each other, suggesting that they are separate constructs?

7. I recommend integrating your findings within your text that you’ve already included under the subheading “The cultural context for paternal involvement and family functioning in Lebanon.” This will make the implications of your findings clearer for the reader.

8. How do you think the fact that tapes were translated into English and then coded impacts your mother-child synchrony measures?

9. The interviews with fathers (page 25) sound very interesting, but I am concerned that they are outside of the scope of the present study, which did not assess barriers to or predictors of paternal involvement.

10. In the final paragraph, the relation between paternal involvement and maternal outcomes/parenting are not discussed as a “call to action” which seems relevant for another future direction (or even integrated into your second call to action).

2. Statistical analyses

11. My biggest concern about the analyses is the relatively large number of tests being done with such a small sample size. However, I think including this concern as a limitation was an appropriate way to handle this for this case as this paper covers a very important topic that likely is challenging to study on a larger scale.

12. Always present both a count and percentage in the text. For example, for child sex and place of mothers’ birth on page 10, paragraph 1.

13. Please include the method by which you assessed model fit in the “Data analysis” section.

14. Please include the standard deviation of child age in the text (page 10, paragraph 1).

15. Were there any significant differences between the participants missing data and those will full data?

16. Did you test for significant differences in indirect effects for maternal distress vs. well-being on both outcomes (i.e., for parenting stress and harsh discipline)? In other words, were the indirect effects for maternal distress significantly stronger for maternal distress compared to well-being?

17. Why did you choose to assess correlations for the dyadic interaction task, rather than conduct path analyses?

3. Clarity of writing

Abstract

18. If the journal guidelines permit, the abstract would benefit from adding the average age and/or age range of participating children.

Introduction

19. The introduction clearly walks the reader through the importance of studying fathers in the context of human-made disasters. I especially appreciated the succinct overview of models of father involvement and the discussion of contextual factors relevant to this population.

20. When you refer to paternal involvement as a “potential modifier” (page 5, paragraph 2 and again on page 7, paragraph 2), it sounds like a moderator. However, based on the articles you have cited and the way the rest of the sentence is written, I believe you are referring to a mediator. If so, I recommend changing “modifier” to “mechanism” or “mediator.”

21. This may come up later, but I found myself wondering what parenting practices in the community would look like (in reference to the operationalization of including involvement in the community on page 5, paragraph 3). The justification for including community involvement is adequate, but an example would be helpful to illustrate what that means. Are you referring to the types of connectedness discussed in the subsequent paragraph?

22. In general, when referring to findings from previous studies, it is better to discuss them in the past tense so as to not over-extend them. For example, on page 6, paragraph 2, you may consider your discussion of the Syrian study to: “A recent study of displaced Syrians in Lebanon showed that self-settled Syrians capitalized on social networks (e.g., by evaluating community opportunities and resources available to them) to create a new livelihood system for themselves, which served as a way to cope with experiences of displacement”

23. In the Objectives subsection, extensive new literature about food insecurity was introduced. I would recommend moving this literature to earlier in the Introduction such that no new literature is introduced when outlining the hypotheses.

24. Minor text edits

a. Page 4, paragraph 2: Change “improved children’s developmental outcomes” to “children’s improved developmental outcomes”

b. Page 5, paragraph 1: In the first sentence, change “exception” to “exceptions”

c. Page 5, paragraph 1: Change “lower levels of maternal distress and higher levels of marital satisfaction, and overall happiness” to “lower levels of maternal distress, higher levels of marital satisfaction, and overall happiness”

d. Page 5, paragraph 2: Change “These models are crucial as they outline the ways in which fathering may influence child development as well as the quality of family relationships” to “These models are crucial as they outline the ways in which fathering may influence child development and the quality of family relationships”

e. Page 8, paragraph 2: Change “Thus, we expected that higher levels of social support” to “Thus, we expected higher levels of social support”

f. Page 9, paragraph 1: Change “household crowing would be association with increased levels of distress and lower levels of well-being” to “household crowing would be associated with increased levels of distress and lower levels of well-being”

Methods

25. Strengths of the Methods section include the thorough descriptions of the measures and the succinct but clear description of data collection procedures.

26. On page 12, paragraph 2: Change “comprised of” to either “composed of” or “comprised”

Results

27. I really appreciate how Figure 1 is formatted. You managed to present a lot of data in a very clear manner.

28. In addition to stating the direction of relations, I recommend stating the strength of the magnitude of associations when discussing correlations in the text.

29. I found myself having to scroll back to the Introduction to review the hypotheses when reading through the results. If you have enough space to add more text, I recommend briefly reiterating the hypotheses when discussing the results.

Discussion

30. Most of my points about the discussion are at the beginning of my comments (in relation to the first criteria).

31. Please consider reiterating the hypotheses and using them as a guide to your Discussion section.

Reviewer #2: This manuscript has several strengths. It is focused on a population that is relatively little studied. The analyses are, for the most part, sophisticated, and the writing is clear.

My greatest concern about this article is about the second set of analyses involving mother-child interactions: they are not compelling, especially when contrasted with the analyses described before them. More specifically, the first set of analyses involves path models including multiple predictors and consideration of both direct and indirect associations. The second set of analyses involves just simple correlations. Of greater concern is the fact that of the 40 values presented in Table 2, only 4 are statistically significant; this is not even what would have been expected based on chance. The authors might consider leaving out these correlations and sticking to just the first set of analyses (also, it is not clear why these correlational findings are discussed before the path models in the Abstract).

Whether or not they retain the correlational analyses noted above, I have some suggestions for the authors to consider as they refine this work for publication. First, they would do well to consider recent publications that directly address some of the points they are emphasizing. Commenting on a collection of articles, for example, Luthar & Eisenberg (2017) indicated that not only is children’s well-being fundamentally tied to the well-being of the mother (usually the primary caregiver), but also that mothers’ well-being versus stress/distress rests fundamentally on the supports that they regularly receive in their everyday lives; naturally, this includes support from their partners and families. This central point is also discussed at length in the recent report from the National Academies of Science, Engineering and Medicine, entitled “Vibrant and Healthy Kids”, as are distinctions between mothers and fathers as primary caregivers.

In discussing their own findings on links between social support and mothers’ well-being, it is not entirely clear why these are described as “weakly related”. The path in question is statistically significant; another way of describing the findings might be that, “Even after taking into account the strong link between perceived paternal involvement and maternal well-being, the link between social support and mothers’ well-being was significant”. From the standpoint of those women who do not have highly involved husbands, this would suggest that high social support could serve some compensatory functions.

Finally, I have some questions about the culture specificity of the findings. On page 21, there is an extensive discussion on family-related norms in the Middle East. However, I am not sure why this statement, for example, might not apply to families generally, regardless of their culture: “…paternal involvement in childrearing, in the mother-child relationship, and in the community may be interpreted by the mother, as a demonstration of sympathy and of her commitment to and bond with the family unit, which could positively affect her well-being.”

6. PLOS authors have the option to publish the peer review history of their article (what does this mean?). If published, this will include your full peer review and any attached files.

Reviewer #1: No

Reviewer #2: No

---

## [Author Response · Author response to Decision Letter 0]

28 Dec 2019

Journal Requirements:

Response: We are now following PLOS ONE’s style requirements according to the style templates provided.

2. Please provide additional details regarding participant consent. In the ethics statement in the Methods and online submission information, please ensure that you have specified whether consent was written or verbal/oral. If consent was verbal/oral, please specify: 1) whether the ethics committee approved the verbal/oral consent procedure, 2) why written consent could not be obtained, and 3) how verbal/oral consent was recorded. If your study included minors, please state whether you obtained consent from parents or guardians in these cases.

Response: We included the following information regarding participant consent: “Recruitment to the study was conducted in close partnership with the community centers, and included sending letters, holding community gatherings and town hall meetings, and following up as needed via telephone calls. The Principal Investigator in Lebanon (GI) obtained written consent from center directors to ensure proper understanding of the study’s purpose, procedures, rights and benefits to all participants. Interested mothers who were eligible to participate were invited to an introductory group meeting where they were scheduled for their initial assessment (baseline), at which time written consent and permission for the children to participate in the study were obtained. Upon consultation with local partners, monetary compensation for each data collection time point was deemed the appropriate incentive. This trial was conducted according to the guidelines of the Human Subjects Committees at Yale University and the Arts, Sciences, and Technology University in Lebanon, who approved all recruitment, assessment, data management, and participant protection procedures. No adverse effects were reported during the study.”

3. We noted in your submission details that a portion of your manuscript may have been presented or published elsewhere. [In this paper, we analyzed baseline data from 104 mother-child dyads who participated in a randomized controlled trial aimed at evaluating the impact of the Mother-Child Education Program on vulnerable populations in Beirut, Lebanon. We used baseline and endpoint data from the same trial to write a paper on the outcome evaluation of a randomized controlled trial concerning the effects of the Mother-Child Education Program on parenting stress and disciplinary practices among refugee and marginalized communities in Lebanon (considered for publication and currently under review in Journal of the American Academy of Child & Adolescent Psychiatry). Moreover, we conducted a process evaluation of the intervention (accepted for publication in the journal New Directions for Child and Adolescent Development). Both manuscripts are attached to this submission as supplemental material.]

* Please clarify whether this publication was peer-reviewed and formally published. If this work was previously peer-reviewed and published, in the cover letter please provide the reason that this work does not constitute dual publication and should be included in the current manuscript.

Response: Both publications were peer-reviewed. The first paper (Ponguta et al., Journal of the American Academy of Child & Adolescent Psychiatry, JAACAP) has been conditionally accepted and is pending a final review of the revisions by the journal. The current manuscript does not constitute dual publication as we focused on associations between paternal involvement and maternal well-being rather than the analysis of program effects on maternal and child outcomes (as reported in the paper by Ponguta et al., JAACAP). 

The other paper (Ponguta et al., in the journal New Directions for Child and Adolescent Development, NDCAD; https://onlinelibrary.wiley.com/doi/10.1002/cad.20314) has been formally published. In this paper, we report findings on key enablers and barriers of participation in the Mother-Child Education Program in Beirut, Lebanon. Therefore, the paper published in NDCAD focuses on a different set of research questions and analytic approach as the manuscript currently under review in PLOS ONE. 

Response: We included a section for the supporting information at the end of the manuscripts and provided the captions of both supporting tables according to PLOE ONE’s style guidelines. 

Reviewer #1: This manuscript covers an important and interesting topic in an often-overlooked population – relations between paternal involvement, contextual factors, and maternal parenting/outcomes in a sample of refugee and socioeconomically disadvantaged families. Please see my comments below for suggestions for improving the manuscript for publication.

1. Conclusions from the data

1. The authors appear to have used face-valid and relatively reliable measures (given the limitations of researching this population) and conducted appropriate analyses. However, their interpretation of their findings in the Discussion would benefit from more explicit integration of their findings in the context of research and practice.

Response: Thank you for this comment. In the course of addressing other pertinent reviewer comments, we have expended the discussion to specifically address broader implications for practice. Specifically, we revised the following section to address the need for further research:

“More research is also needed on the validation of the measures among disadvantaged and refugee families in Lebanon. For instance, there may be distinct profiles of psychological distress that may be more susceptible to the qualities of the mother-father relationship than others. Finally, future research should expand on the model presented here by examining the impact of paternal involvement on child development outcomes via maternal well-being and parenting behaviors.”

We also revised the conclusions to emphasize our findings in the context of practice in fragile and humanitarian settings:

“In conclusion, we would like to offer two calls to action. First, research, monitoring, and evaluation in humanitarian settings should pay greater attention to co-developing robust, context-specific, and time-specific tools that capture paternal involvement and the relation with maternal well-being and parenting together with families living in these areas. Second, after such tools have been developed and tested, there should be targeted efforts to translate the knowledge gained into evidence-based capacity building and actionable recommendations for practitioners. First and foremost, such an initiative would contribute to the discourse about the need for paternal involvement interventions in humanitarian settings. The translation of such knowledge should occur in collaboration with the families and communities themselves.”

2. Although there were no direct relations between paternal involvement and maternal harsh discipline, there were indirect relations. The Discussion would be strengthened by discussing these pathways and their implications for research and practice.

Response: Thank you very much for this suggestion. We included the following text at the beginning of the Discussion section to address the reviewer’s comment:

“Thus, although there were no direct relations between paternal involvement and maternal harsh discipline and parenting stress, there were indirect relations via maternal perceptions of distress. The negative association between paternal involvement and maternal psychological distress (e.g., maternal depressive symptoms), has been well-documented in research on the influence of the relations with nonresident fathers on the well-being of mothers (specifically among non-employed mothers; 1), perceptions of support from mothers’ partners who are not formally identified as stepfathers (e.g., among Mexican-origin mothers with 3-year-old children; 2), and the contributions of the quality of the marital relationship (e.g., the extent of agreement between partners on a range of family-related issues) to maternal well-being among mothers of children with developmental disabilities (3). Similar to the findings of the present study, maternal distress and parenting stress are often interrelated (4-6), whereas evidence supporting a direct link of paternal involvement/support and perceptions of parenting/parental stress is less robust (1, 2).”

3. How do you propose that maternal gatekeeping relates to paternal involvement and subsequent maternal harsh discipline?

Response: In our previous manuscript, we stated that it is important to consider aspects related to maternal gatekeeping when attempting to assess the fathers’ perceived involvement, as studies have found that mothers’ beliefs about the role of fathers regulate the relationship between fathers’ perceived roles and actual levels of involvement (7). 

We expanded on the notion of maternal gatekeeping in the revised manuscript by including the following details: 

“In their seminal research on maternal gatekeeping, Allen and Hawkins (8) conceptualized maternal gatekeeping as three interrelated perceptions and behaviors: (a) mothers' reluctance to relinquish responsibility over family matters by setting rigid standards; (b) external validation of a mothering identity; and (c) differentiated conceptions of family roles. More recently, Puhlman and Pasley (9, 10) proposed a three-dimensional model of gatekeeping comprised of encouragement (i.e., facilitative dimension; gate open), discouragement (i.e., inhibitory dimension; gate closed) and control. Encouragement is characterized by positive feedback (e.g., expression of positive regard through compliments) and welcoming behaviors of the mothers, which support father involvement with children. Discouragement, on the other hand, entails maternal expression of negativity toward fathers through behaviors such as frequent criticism and avoidance of co-parenting interactions. 

Collectively, research points to standard and responsibility, maternal identity confirmation, differentiated family roles, encouragement, discouragement, and control as crucial dimensions of maternal gatekeeping. Additional research has examined antecedents of maternal gatekeeping and father involvement and highlighted the differential roles of fathering identity (i.e., the salience of the role as a father) and maternal influences (e.g., psychological correlates such as personality, negative emotionality, self-esteem) as well as life transitions and social context (e.g., the socially scripted role of the mother), among other factors (7, 12, 13). In line with this previous research, mothers in our study who reported high levels of well-being and low levels of distress may be more likely to encourage fathers to take responsibility for family matters. Conversely, low levels of well-being and high levels of distress may facilitate inhibitory gatekeeping behaviors such as discouragement as a way to gain appreciation from one’s social context(s). Together, these assumptions point to complex, dynamic and reciprocal factors that, to our knowledge, have not been examined in relation to experiences of displacement and migration. Drawing from recent research on the associations between supportive relationships and caregiver well-being (14), future research may seek to examine the role of caregiver well-being in curtailing maternal gatekeeping and promoting cooperation among caregivers in general (15), and in refugee and marginalized communities in particular.”

4. The statement that “fathers may provide a source of support that affects positive interactions between the mother and her child” were not supported by the data because indirect relations between paternal involvement and maternal well-being, via mother-child synchrony, were not assessed.

Response: This conclusion was based on the (small) correlations between paternal involvement and the observed mother-child interactions. We did not aim to include mother-child interactions in the structural equation model due to the modest sample size and to avoid noise in the analysis due to different data collection methods. To address this valuable comment, we added the following text to the Discussion to call for future studies that examine potential mediators of the relations between paternal involvement and mother-child interactions: 

“The latter conclusion was supported, in part, by the (small) correlations between paternal involvement and maternal and child behavior during the interaction task. However, maternal well-being was not significantly associated with the indicators of observed mother-child interactions. Thus, future studies should examine the characteristics and factors that may mediate the association between paternal involvement and mother-child interactions.”

5. The Discussion would be strengthened by the inclusion of your findings on relations between basic needs and maternal outcomes. What are the implications of your findings?

Response: Many thanks for this recommendation. We agree that discussion should more explicitly include the implications of the association found between basic needs and distress. To address this comment, we added the following text and relevant literature:

“We drew from the existing literature in relatively comparable populations, and our data indicated that a composite assessing food and human insecurity and household crowding index, was a valid proxy to assess a mother’s ability to meet her/her family’s basic needs. As expected, our findings indicated that an inability to meet the family’s basic needs was associated with distress among mothers. These findings are consistent with a substantive body of literature linking socioeconomic need and maternal distress during infant (16) and through early childhood development (17). Importantly, maternal distress was associated with parenting stress as well as harsh discipline in our sample, suggesting that structural challenges are related to parental functioning and behaviors. These findings implicate economic and food security supports as key strategies to accompany targeted social interventions.”

6. Please elaborate about the role of perceived social support (page 20, last paragraph). Is your argument that both perceived social support and paternal involvement were separately related to maternal outcomes, but not strongly related to each other, suggesting that they are separate constructs?

Response: Thank you very much for this comment. We are not concluding that perceived social support and paternal involvement are separate constructs as we have not conducted an in-depth psychometric evaluation (e.g., construct validity) and comparison of both instruments. We added the following text in the Discussion to emphasize the roles of perceived social support and paternal involvement for maternal outcomes: 

“Findings from the path model showed that paternal involvement was related to both maternal distress and maternal well-being when controlling for basic needs and perceptions of social support. Whereas, maternal ratings of perceived social support were only related to maternal well-being but not to maternal distress when controlling for basic needs and paternal involvement. These findings support the notion that both constructs may function as distinct sources to promote maternal well-being. Moreover, paternal involvement (but not perceived social support) may alleviate the stress the mother experiences in being a parent in under-resourced or at-risk circumstances.”

7. I recommend integrating your findings within your text that you’ve already included under the subheading “The cultural context for paternal involvement and family functioning in Lebanon.” This will make the implications of your findings clearer for the reader.

Response: We are now referring to key findings in the first paragraph in the section on “The cultural context for paternal involvement and family functioning in Lebanon” to substantiate our interpretations. Regarding a related conclusion, we also included the statement that “future studies should examine the extent and nature of additional resources available to the family due to increased paternal involvement, and whether these resources positively impact maternal well-being.”

8. How do you think the fact that tapes were translated into English and then coded impacts your mother-child synchrony measures?

Response: We added the information that the videotaped interactions were translated and subtitled by a native Arabic speaker with relevant credentials (M.D., M.Sc.). We followed standard coding protocols and procedures outlined in the cited publications and manuals. By doing so, we assumed that the translation of videotaped interactions would introduce less bias as the next best alternative, that is, the translation of all coding manuals into Arabic and the subsequent training of Arabic-speaking staff. We added the possibility of bias due to translating the videotaped interactions from Arabic into English as a limitation of the study. We also added that “Future studies should aim to validate the coding protocols and procedures with Arabic-speaking populations in similar contexts.”

9. The interviews with fathers (page 25) sound very interesting, but I am concerned that they are outside of the scope of the present study, which did not assess barriers to or predictors of paternal involvement.

Response: We consider it important to contextualize the (quantitative) findings of the study by including some of the key findings from the interviews with fathers in these communities. Paternal involvement is a key variable in our analytic model. The interviews point to additional anecdotal observations about potential factors contributing to paternal involvement that should be the focus of future studies. We revised the paragraph in the section “Future directions and conclusions” to better express this idea.

10. In the final paragraph, the relation between paternal involvement and maternal outcomes/parenting are not discussed as a “call to action” which seems relevant for another future direction (or even integrated into your second call to action).

Response: Thank you for pointing this out. We revised the statements accordingly to emphasize the need to examine relations between paternal involvement, maternal well-being, and parenting. The second “call to action” was meant to contribute to the discourse about the need for paternal involvement interventions in humanitarian settings. We have added a sentence to clarify our intentions. 

2. Statistical analyses

11. My biggest concern about the analyses is the relatively large number of tests being done with such a small sample size. However, I think including this concern as a limitation was an appropriate way to handle this for this case as this paper covers a very important topic that likely is challenging to study on a larger scale.

Response: Thank you for this suggestion. We added the following text to the Limitations: “Moreover, a relatively large number of tests were conducted based on this small sample. Thus, the reported levels of statistical significance should be reported with caution.”

12. Always present both a count and percentage in the text. For example, for child sex and place of mothers’ birth on page 10, paragraph 1.

Response: We have revised the Sample description accordingly. 

13. Please include the method by which you assessed model fit in the “Data analysis” section.

Response: We added the following text to the section “Data analysis”: The overall goodness-of-fit of the path model was evaluated based on the following fit indices: the comparative fit index (CFI), the standardized root mean square residual (SRMR), and the root-mean-square error of approximation (RMSEA). Estimates of the CFI close to .95, SRMR close to .08, and RMSEA close to .06 indicate adequate model fit (Hu & Bentler, 1999).

14. Please include the standard deviation of child age in the text (page 10, paragraph 1).

Response: This information is now included in the text.

15. Were there any significant differences between the participants missing data and those will full data?

Response: Little’s MCAR test indicates that data were missing completely at random, �2 (121) = 113.50, p = .67. We added this information to the Results section.

16. Did you test for significant differences in indirect effects for maternal distress vs. well-being on both outcomes (i.e., for parenting stress and harsh discipline)? In other words, were the indirect effects for maternal distress significantly stronger for maternal distress compared to well-being?

Response: Thank you for this suggestion! We re-examined our model to compare the estimated indirect effects. We tested four differences: 

(1) the difference in the indirect effect of paternal involvement on maternal harsh discipline via maternal distress and the indirect effect of paternal involvement on parenting stress via maternal distress; 

(2) the difference in the indirect effect of paternal involvement on maternal harsh discipline via maternal well-being and the indirect effect of paternal involvement on parenting stress via maternal well-being; 

(3) the difference in the indirect effect of paternal involvement on maternal harsh discipline via maternal distress and the indirect effect of paternal involvement on harsh discipline via maternal well-being; and 

(4) the difference in the indirect effect of paternal involvement on parenting stress via maternal distress and the indirect effect of paternal involvement on parenting stress via maternal well-being.

Model results showed that the indirect effect of paternal involvement on parenting stress via maternal distress was stronger compared to the indirect effect of paternal involvement on harsh discipline via maternal distress, B = 0.29, SE = .131, p = .027). The differences in other indirect effects of paternal involvement on harsh discipline and parenting stress via maternal distress and well-being were small and statistically non-significant. We added these findings to the Results section.

17. Why did you choose to assess correlations for the dyadic interaction task, rather than conduct path analyses?

Response: As mentioned in response to comment 4, we did not aim to include mother-child interactions in the structural equation model due to the modest sample size and to avoid noise in the analysis due to different data collection methods. We coded a total of 8 indicators of mother-child interactions (i.e., sensitivity, unobtrusive behavior, disengagement, positive regard, synchrony, child negative affect, child non-compliance, and child positive affect). Including these variables in the path analyses would add significant computational complexity, even if we were able to compute composite scores (e.g., one variable capturing maternal behaviors, one variable for mother-child synchrony, and one variable for child behavior/affect). Therefore, we decided to report zero-order correlations and report the findings with a focus on the correlations between paternal involvement and indicators of mother-child interactions. 

3. Clarity of writing

Abstract

18. If the journal guidelines permit, the abstract would benefit from adding the average age and/or age range of participating children.

Response: We added this information to the Abstract.

Introduction

19. The introduction clearly walks the reader through the importance of studying fathers in the context of human-made disasters. I especially appreciated the succinct overview of models of father involvement and the discussion of contextual factors relevant to this population.

Response: We appreciate your assessment of the introduction. 

20. When you refer to paternal involvement as a “potential modifier” (page 5, paragraph 2 and again on page 7, paragraph 2), it sounds like a moderator. However, based on the articles you have cited and the way the rest of the sentence is written, I believe you are referring to a mediator. If so, I recommend changing “modifier” to “mechanism” or “mediator.”

Response: On page 5, we adopted the term “modifier” from Barker, Iles, and Ramchandani (2017; cited in the text). In Figure 2 of their paper, the authors list “father’s involvement” as an “effect modifier”, with arrows pointing at the associations between (1) paternal psychopathology and three “proposed environmental mechanisms” (i.e., increased couple conflict, impaired parenting, and effect on maternal psychological health); and between (2) the three “proposed environmental mechanisms” and child psychopathology. The figure implies that the father’s involvement is hypothesized to moderate the proposed pathways. We prefer to keep the term “modifier” on page 5 to be consistent with the cited reference. On page 7, we have changed “modify” to “mitigate” to avoid confusion with the conceptualization of moderator and mediator variables in statistics. 

21. This may come up later, but I found myself wondering what parenting practices in the community would look like (in reference to the operationalization of including involvement in the community on page 5, paragraph 3). The justification for including community involvement is adequate, but an example would be helpful to illustrate what that means. Are you referring to the types of connectedness discussed in the subsequent paragraph?

Response: We added the following text to the Introduction to provide some examples of paternal involvement in the community: “Examples of paternal involvement in the community assessed in the present study are the extent to which the father attends community events, spends time with other fathers in the community, and enjoys spending time with other families.”

22. In general, when referring to findings from previous studies, it is better to discuss them in the past tense so as to not over-extend them. For example, on page 6, paragraph 2, you may consider your discussion of the Syrian study to: “A recent study of displaced Syrians in Lebanon showed that self-settled Syrians capitalized on social networks (e.g., by evaluating community opportunities and resources available to them) to create a new livelihood system for themselves, which served as a way to cope with experiences of displacement”.

Response: We revised the manuscript to use past tense (where appropriate). 

23. In the Objectives subsection, extensive new literature about food insecurity was introduced. I would recommend moving this literature to earlier in the Introduction such that no new literature is introduced when outlining the hypotheses.

Response: We moved the section about food insecurity to the second to last paragraph on in the Introduction.

24. Minor text edits

a. Page 4, paragraph 2: Change “improved children’s developmental outcomes” to “children’s improved developmental outcomes”

b. Page 5, paragraph 1: In the first sentence, change “exception” to “exceptions”

c. Page 5, paragraph 1: Change “lower levels of maternal distress and higher levels of marital satisfaction, and overall happiness” to “lower levels of maternal distress, higher levels of marital satisfaction, and overall happiness”

d. Page 5, paragraph 2: Change “These models are crucial as they outline the ways in which fathering may influence child development as well as the quality of family relationships” to “These models are crucial as they outline the ways in which fathering may influence child development and the quality of family relationships”

e. Page 8, paragraph 2: Change “Thus, we expected that higher levels of social support” to “Thus, we expected higher levels of social support”

f. Page 9, paragraph 1: Change “household crowing would be association with increased levels of distress and lower levels of well-being” to “household crowing would be associated with increased levels of distress and lower levels of well-being”

Response: Thank you for catching these typos and grammatical errors. We revised the text accordingly. 

Methods

25. Strengths of the Methods section include the thorough descriptions of the measures and the succinct but clear description of data collection procedures.

Response: Thank you!

26. On page 12, paragraph 2: Change “comprised of” to either “composed of” or “comprised”

Response: We changed the wording to “composed of”.

Results

27. I really appreciate how Figure 1 is formatted. You managed to present a lot of data in a very clear manner.

Response: Thank you very much!

28. In addition to stating the direction of relations, I recommend stating the strength of the magnitude of associations when discussing correlations in the text.

Response: We revised the first paragraph of the Results section to indicate the magnitude (i.e., small, medium) of the reported correlations.

29. I found myself having to scroll back to the Introduction to review the hypotheses when reading through the results. If you have enough space to add more text, I recommend briefly reiterating the hypotheses when discussing the results.

Response: We revised the Results section to state that the correlations support our initial hypotheses. 

Discussion

30. Most of my points about the discussion are at the beginning of my comments (in relation to the first criteria).

Response: We reviewed them carefully and hope that we addressed them adequately.

31. Please consider reiterating the hypotheses and using them as a guide to your Discussion section.

Response: We paraphrased our hypotheses and expectations and added them to the Discussion to facilitate the flow of the text.

 

Reviewer #2: This manuscript has several strengths. It is focused on a population that is relatively little studied. The analyses are, for the most part, sophisticated, and the writing is clear.

Comment 1: My greatest concern about this article is about the second set of analyses involving mother-child interactions: they are not compelling, especially when contrasted with the analyses described before them. More specifically, the first set of analyses involves path models including multiple predictors and consideration of both direct and indirect associations. The second set of analyses involves just simple correlations. Of greater concern is the fact that of the 40 values presented in Table 2, only 4 are statistically significant; this is not even what would have been expected based on chance. The authors might consider leaving out these correlations and sticking to just the first set of analyses (also, it is not clear why these correlational findings are discussed before the path models in the Abstract).

Response: As mentioned in response to reviewer 1, we did not aim to include mother-child interactions in the structural equation model due to the modest sample size and to avoid noise in the analysis due to different data collection methods. We coded a total of 8 indicators of mother-child interactions (i.e., sensitivity, unobtrusive behavior, disengagement, positive regard, synchrony, child negative affect, child non-compliance, and child positive affect). Including these variables in the path analyses would add significant computational complexity, even if we were able to compute composite scores (e.g., one variable capturing maternal behaviors, one variable for mother-child synchrony, and one variable for child behavior/affect). Therefore, we decided to report zero-order correlations and report the findings with a focus on the correlations between paternal involvement and indicators of mother-child interactions.

As per our initial hypothesis, “we expected to find positive associations among paternal involvement, indicators of the mother-child interaction (see Methods) and maternal well-being, and negative associations with regard to maternal distress (18).” The additional correlations between maternal outcomes and mother-child interactions reported in Table 2 were not the focus of the present study and were not based on hypotheses. As the focus of this study was on paternal involvement, we decided to report only the correlations between paternal involvement and the mother-child interactions. We removed the correlations from the table and added “data not shown” after the following sentence: “Correlations between the indicators of mother-child interaction and mothers’ self-reported well-being and parenting were overall small and statistically non-significant (data not shown).”

Comment 2: Whether or not they retain the correlational analyses noted above, I have some suggestions for the authors to consider as they refine this work for publication. First, they would do well to consider recent publications that directly address some of the points they are emphasizing. Commenting on a collection of articles, for example, Luthar & Eisenberg (2017) indicated that not only is children’s well-being fundamentally tied to the well-being of the mother (usually the primary caregiver), but also that mothers’ well-being versus stress/distress rests fundamentally on the supports that they regularly receive in their everyday lives; naturally, this includes support from their partners and families. This central point is also discussed at length in the recent report from the National Academies of Science, Engineering and Medicine, entitled “Vibrant and Healthy Kids”, as are distinctions between mothers and fathers as primary caregivers.

Response: Thank you for these valuable suggestions. We added the following information in the Discussion section: “It is clear that a child’s well-being is fundamentally tied to the well-being of their primary caregivers, typically their mothers, and that the mothers’ well-being rests fundamentally on the supports (e.g., psychological, emotional support) that they receive from people in their everyday lives as well as the quality of the relationship (e.g., the level of conflict, cooperation) between the primary caregivers (e.g., 14). This reality is also discussed at length in the recent comprehensive report from the National Academies of Science, Engineering, and Medicine, entitled “Vibrant and Healthy Kids” (19). For instance, the authors of the report recommend that “special attention should be given to the recruitment, retention, and support of men and fathers parenting young children from underserved populations.” (p. 9). We would like to underscore this recommendation, particularly with regard to fathers from refugee and marginalized communities.”

Comment 3: In discussing their own findings on links between social support and mothers’ well-being, it is not entirely clear why these are described as “weakly related”. The path in question is statistically significant; another way of describing the findings might be that, “Even after taking into account the strong link between perceived paternal involvement and maternal well-being, the link between social support and mothers’ well-being was significant”. From the standpoint of those women who do not have highly involved husbands, this would suggest that high social support could serve some compensatory functions.

Response: Thank you for this suggestion. We added the following text to reflect this idea: “Even after taking into account the strong link between perceived paternal involvement and maternal well-being, the associations between social support and mothers’ well-being was small but statistically significant. Maternal perceptions of social support, however, were not significantly related to ratings of distress.”

Comment 4: Finally, I have some questions about the culture specificity of the findings. On page 21, there is an extensive discussion on family-related norms in the Middle East. However, I am not sure why this statement, for example, might not apply to families generally, regardless of their culture: “…paternal involvement in childrearing, in the mother-child relationship, and in the community may be interpreted by the mother, as a demonstration of sympathy and of her commitment to and bond with the family unit, which could positively affect her well-being.”

Response: Thank you for this valuable comment. We added “Not unique to families in the Middle East” to the sentence to reflect that this idea applied to families generally. We also added the following sentence to tie the conclusion to our study: “In our data, this conclusion is, in part, supported by the finding that paternal involvement was positively related to maternal well-being.” We also (re-)stated our findings after the sentence starting with “Conversely, …” to support the conclusion with data from our study. 

Furthermore, we added the following information to address the reviewer’s comment about the culture specificity (or the lack thereof): “In this regard, we need to be careful not to generalize the findings across culture as the literature on cross-cultural co-parenting is scarce, and recent reviews have highlighted the need for additional research globally on paternal engagement in parenting interventions (20).”

 

References

1. Jackson AP. The effects of nonresident father involvement on single black mothers and their young children. Social Work. 1999;44(2):156-66.

2. Gonzalez H, Barnett MA. Romantic partner and biological fahter support: Associations with maternal distress in low-income Mexican-origin families. Family Relations. 2014;63:371-83.

3. Kersh J, Hedvat TT, Hauser-Cram P, Warfield ME. The contribution of marital quality to the well-being of parents of children with developmental disabilities. Journal of Intellectual Disability Research. 2006;50(12):883-93.

4. Goodman JH. Influences of maternal postpartum depression on fathers and on father-infant interactions. Infant Mental Health Journal. 2008;29(6):624-43.

5. Misri S, Kendrick K, Overlander TF, Norris S, Tomfohr L, Zhang H, et al. Antenatal depression and anxiety affect postpartum parenting stress: A longitudinal, prospective study. Canadian Journal of Psychiatry. 2010;55(4):222-8.

6. Thomason E, Volling BL, Flynn HA, McDonough SC, Marcus SM, Lopez JF, et al. Parenting stress and depressive symptoms in postpartum mothers: Bidirectional or unidirectional effects? Infant Behavior and Development. 2014;37(3):406-15.

7. McBride BA, Brown GL, Bost KK, Shin N, Vaughn B, Korth B. Paternal identity, maternal gatekeeping, and father involvement. Family Relations. 2005;54:360-72.

8. Allen SM, Hawkins AJ. Maternal gatekeeping: Mothers’ beliefs and behaviors that inhibit greater father involvement in family work. Journal of Marriage & Family. 1999;61:199-212.

9. Puhlman DJ, Pasley K. The Maternal Gatekeeping Scale: Constructing a Measure. Family Relations. 2017;66(5):824-38.

10. Puhlman DJ, Pasley K. Rethinking maternal gatekeeping. Journal of Family Theory and Review. 2013;5:176-93.

11. Schoppe-Sullivan SJ, Brown GL, Cannon EA, Mangelsdorf SC, Sokolowski MS. Maternal gatekeeping, coparenting quality, and fathering behavior in families with infants. Journal of Family Psychology. 2008;22(3):389-98.

12. Cannon EA, Schoppe-Sullivan SJ, Mangelsdorf SC, Brown GL, Szewczyk Sokolowski M. Parent characteristics as antecedents of maternal gatekeeping and fathering behavior. Family Process. 2008;47:501-19.

13. Gaunt R. Maternal gatekeeping: Antecedents and consequences. Journal of Family Issues. 2008;29(3):373-95.

14. Luthar SS, Eisenberg N. Resilient Adaptation Among At-Risk Children: Harnessing Science Toward Maximizing Salutary Environments. Child Dev. 2017;88(2):337-49.

15. Pruett MK, Pruett KD, Cowan CP, Cowan PA. Enhancing Father Involvement in Low-Income Families: A Couples Group Approach to Preventive Intervention. Child development. 2017;88(2):398-407.

16. Goyal D, Gay C, Lee KA. How much does low socioeconomic status increase the risk of prenatal and postpartum depressive symptoms in first-time mothers? Womens Health Issues. 2010;20(2):96-104.

17. Steele H, Bate J, Steele M, Dube SR, Danskin K, Knafo H, et al. Adverse childhood experiences, poverty, and parenting stress. Canadian Journal of Behavioural Science / Revue canadienne des sciences du comportement. 2016;48(1):32-8.

18. Fletcher R. Promoting infant well-being in the context of maternal depression by supporting the father. Infant Mental Health Journal. 2009;30(1):95-102.

19. National Academies of Sciences E, and Medicine. Vibrant and Healthy Kids: Aligning Science, Practice, and Policy to Advance Health Equity. Washington, DC: The National Academies Press; 2019.

20. Panter-Brick C, Burgess A, Eggerman M, McAllister F, Pruett K, Leckman JF. Practitioner review: Engaging fathers--recommendations for a game change in parenting interventions based on a systematic review of the global evidence. Journal of Child Psychology and Psychiatry. 2014;55(11):1187-212.

---

## [Decision Letter · Decision Letter 1]

13 Jan 2020

PONE-D-19-24487R1

Maternal perceptions of father involvement among refugee and disadvantaged families in Beirut, Lebanon

PLOS ONE

Dear Dr. Hein,

Thank you for submitting your manuscript to PLOS ONE. After careful consideration, we feel that it has merit but does not fully meet PLOS ONE’s publication criteria as it currently stands. Therefore, we invite you to submit a revised version of the manuscript that addresses the points raised during the review process.

We would appreciate receiving your revised manuscript by Feb 27 2020 11:59PM. To enhance the reproducibility of your results, we recommend that if applicable you deposit your laboratory protocols in protocols.io, where a protocol can be assigned its own identifier (DOI) such that it can be cited independently in the future. For instructions see: http://journals.plos.org/plosone/s/submission-guidelines#loc-laboratory-protocols

We look forward to receiving your revised manuscript.

Kind regards,

Florian Fischer

Academic Editor

PLOS ONE

Reviewers' comments:

Reviewer's Responses to Questions

**Comments to the Author**

1. If the authors have adequately addressed your comments raised in a previous round of review and you feel that this manuscript is now acceptable for publication, you may indicate that here to bypass the “Comments to the Author” section, enter your conflict of interest statement in the “Confidential to Editor” section, and submit your "Accept" recommendation.

Reviewer #1: (No Response)

Reviewer #2: (No Response)

2. Is the manuscript technically sound, and do the data support the conclusions?

Reviewer #1: Yes

Reviewer #2: Partly

3. Has the statistical analysis been performed appropriately and rigorously? 

Reviewer #1: Yes

Reviewer #2: Yes

4. Have the authors made all data underlying the findings in their manuscript fully available?

Reviewer #1: Yes

Reviewer #2: Yes

5. Is the manuscript presented in an intelligible fashion and written in standard English?

Reviewer #1: Yes

Reviewer #2: Yes

6. Review Comments to the Author

Reviewer #1: The authors did an excellent job addressing prior comments and concerns. I only noticed a few typos that I wanted to point out:

- p. 23, para. 1: change "infant" to "infancy"

- p. 23, para. 1: change "as well as" to "and" in the sentence that starts with "Importantly, maternal distress..."

- p. 24, para. 2: change "comprised of" to "composed of"

Reviewer #2: The authors have done a very good job in attending to comments and suggestions previously offered, leading to further strengthening of a manuscript that was already very strong.

I have just two substantive comments to offer as they finalize this work for publication.

The first is that the authors temper the conclusions they offer about what they see as the lack of associations involving mothers’ social support, for a couple of reasons.

First, the fact that a particular path coefficients does not meet the cutoff of p < .05 does not mean that the construct is actually of no importance in this sample (or elsewhere in the real world). Absolute values of coefficients that are interpreted as significant in this study range from as low as .19 to as high as .45. The authors have not conducted significance testing to determine, for example, that coefficients for dotted paths described as non-significant are in fact statistically significant from those coefficients that they have reported as significant (and thus, by implication, of “real world importance”). Thus, I believe it is an over interpretation to conclude, in this statement, that ”paternal involvement (but not perceived social support) may alleviate the stress the mother experiences in being a parent in under-resourced or at-risk circumstances.”

Personally, I would suggest just leaving out the one sentence above, that is currently between the following two, and instead simply state that “These findings support the notion that both constructs may function as distinct sources to promote maternal well-being...... Together, these findings support the notion that maternal distress may mediate the association between maternal perceptions of paternal involvement and both maternal parenting stress and maternal harsh discipline.”

Another issue to consider is that these authors’ findings on social support may have rested partly on measurement issues. It seems that at least 3 of the 7 items on support had to do with frequency of connecting with close others. For a mother with an uninvolved, unsupportive husband, who is sole caregiver of her children, there would be obvious constraints in how much she can visit with others in her daily life.

I would like to emphasize that this is an issue of importance not just for science but as importantly, for what is conveyed to lay folk and policy makers. In other words, I would urge the authors to consider carefully the message that they are disseminating regarding what social supports can, and *cannot* do, for mothers at risk. Indeed, the NASEM (2019) consensus report cited has conclusively stated that the presence of dependable supports is actually the single most important protective factor for all types of moms at risk (and therefore, for their children). For the many single or unhappily married mothers in the world, it would be disheartening indeed to hear the message, from these authoritative authors, that if the child’s father remains uninvolved, other caring relationships in their lives will be of no use to them in terms of bolstering their well-being and parenting.

If the authors do in fact wish to retain this suggestion, then I would very much like to see a conceptual explanation of why their conclusions depart so drastically from those unequivocally stated in the NASEM review.

The second issue concerns these statements on P 31:

“Anecdotal comments from fathers in refugee settings revealed various unique concerns and worries that may prevent fathers from being maximally involved with their family. Common stressors include a feeling of powerlessness due to underemployment and lack of ability to provide for one’s family. Perceived isolation and discrimination from the larger community may also highly affect the fathers’ well-being. Together, these extreme stressors may affect fathers’ well-being more generally, which can compromise their levels of involvement. These anecdotal observations may reveal key issues that need to be targeted in any program. “

I see these comments as in fact providing rich insights into why it is hard for fathers to be more involved with their families. At the same time, I’m left disappointed with the directions offered for future interventions. In other words, would a new family-based program actually reduce the major listed concerns for fathers, such as “underemployment and lack of ability to provide for one’s family (and) perceived isolation and discrimination“? If the authors have suggestions of strategies for how these could realistically be done, it would be wonderful to hear these spelled out. If they do not, it would be useful to acknowledge that these are clearly critical issues, but not easily solved at the level of family-based interventions - they will need changes at larger levels of national policies and perhaps, community norms.

In sum, I congratulate these authors again for a very fine piece of work! I have no doubt that it will make a strong contribution to the literature.

7. PLOS authors have the option to publish the peer review history of their article (what does this mean?). If published, this will include your full peer review and any attached files.

Reviewer #1: No

Reviewer #2: No

---

## [Author Response · Author response to Decision Letter 1]

10 Feb 2020

Reviewer #1: The authors did an excellent job addressing prior comments and concerns. I only noticed a few typos that I wanted to point out:

- p. 23, para. 1: change "infant" to "infancy"

- p. 23, para. 1: change "as well as" to "and" in the sentence that starts with "Importantly, maternal distress..."

- p. 24, para. 2: change "comprised of" to "composed of"

Response: Thank you for pointing out these grammatical issues. All issues have been corrected.

Reviewer #2: The authors have done a very good job in attending to comments and suggestions previously offered, leading to further strengthening of a manuscript that was already very strong.

I have just two substantive comments to offer as they finalize this work for publication.

The first is that the authors temper the conclusions they offer about what they see as the lack of associations involving mothers’ social support, for a couple of reasons.

Comment 1: First, the fact that a particular path coefficients does not meet the cutoff of p < .05 does not mean that the construct is actually of no importance in this sample (or elsewhere in the real world). Absolute values of coefficients that are interpreted as significant in this study range from as low as .19 to as high as .45. The authors have not conducted significance testing to determine, for example, that coefficients for dotted paths described as non-significant are in fact statistically significant from those coefficients that they have reported as significant (and thus, by implication, of “real world importance”). Thus, I believe it is an over interpretation to conclude, in this statement, that ”paternal involvement (but not perceived social support) may alleviate the stress the mother experiences in being a parent in under-resourced or at-risk circumstances.”

Personally, I would suggest just leaving out the one sentence above, that is currently between the following two, and instead simply state that “These findings support the notion that both constructs may function as distinct sources to promote maternal well-being...... Together, these findings support the notion that maternal distress may mediate the association between maternal perceptions of paternal involvement and both maternal parenting stress and maternal harsh discipline.”

Response: We appreciate this comment. Both constructs were indeed associated with maternal well-being. Perceived social support was associated with maternal distress as well, despite to a lesser degree (beta = -0.095) compared to maternal well-being (beta = 0.194). Thus, we agree to take out the sentence as suggested by the reviewer. 

Comment 2: Another issue to consider is that these authors’ findings on social support may have rested partly on measurement issues. It seems that at least 3 of the 7 items on support had to do with frequency of connecting with close others. For a mother with an uninvolved, unsupportive husband, who is sole caregiver of her children, there would be obvious constraints in how much she can visit with others in her daily life.

I would like to emphasize that this is an issue of importance not just for science but as importantly, for what is conveyed to lay folk and policy makers. In other words, I would urge the authors to consider carefully the message that they are disseminating regarding what social supports can, and *cannot* do, for mothers at risk. Indeed, the NASEM (2019) consensus report cited has conclusively stated that the presence of dependable supports is actually the single most important protective factor for all types of moms at risk (and therefore, for their children). For the many single or unhappily married mothers in the world, it would be disheartening indeed to hear the message, from these authoritative authors, that if the child’s father remains uninvolved, other caring relationships in their lives will be of no use to them in terms of bolstering their well-being and parenting.

If the authors do in fact wish to retain this suggestion, then I would very much like to see a conceptual explanation of why their conclusions depart so drastically from those unequivocally stated in the NASEM review.

Response: We carefully reviewed the manuscript (particularly the Discussion section) to make sure we do not (unintentionally) minimize the role of social support in the lives of mothers at risk. As evidenced by the bivariate correlations, perceived social support was positively associated with maternal well-being (r = .27) and negatively associated with maternal distress (r = -.28). 

We added the following text to emphasize the important role of social support: “The findings of our study support this notion. As evidenced by the bivariate correlations, higher levels of perceived social support were associated with higher levels of subjective maternal well-being and lower ratings of maternal distress. Thus, although our focus was on father-mother relations, we fully acknowledge that supportive relations manifest among wider networks.“

Comment 3: The second issue concerns these statements on P 31:

“Anecdotal comments from fathers in refugee settings revealed various unique concerns and worries that may prevent fathers from being maximally involved with their family. Common stressors include a feeling of powerlessness due to underemployment and lack of ability to provide for one’s family. Perceived isolation and discrimination from the larger community may also highly affect the fathers’ well-being. Together, these extreme stressors may affect fathers’ well-being more generally, which can compromise their levels of involvement. These anecdotal observations may reveal key issues that need to be targeted in any program. “

I see these comments as in fact providing rich insights into why it is hard for fathers to be more involved with their families. At the same time, I’m left disappointed with the directions offered for future interventions. In other words, would a new family-based program actually reduce the major listed concerns for fathers, such as “underemployment and lack of ability to provide for one’s family (and) perceived isolation and discrimination“? If the authors have suggestions of strategies for how these could realistically be done, it would be wonderful to hear these spelled out. If they do not, it would be useful to acknowledge that these are clearly critical issues, but not easily solved at the level of family-based interventions - they will need changes at larger levels of national policies and perhaps, community norms. 

In sum, I congratulate these authors again for a very fine piece of work! I have no doubt that it will make a strong contribution to the literature.

Response: Thank you for your thoughtful comment. We agree that strong implications for policy and programming are a desirable outcome of any research. However, we are careful not to overstate the informative value of our data. 

We added the following text to convey our recommendations for future studies aimed at developing policy/program recommendations: “These are critical issues from the perceptive of the fathers we interviewed. However, they are neither solely nor easily solved at the level of family-based intervention. Among many factors, cultural, gender, and migration dynamics, as well as community norms and national policies need to be considered to devise strong policy and/or program recommendations, which are outside the scope of this work. The incremental contribution of this study is to substantiate the role of paternal engagement and support, with further research needed to develop effective programs and strategies that may reduce the major concerns voiced by fathers.”

---

## [Editor Report · Decision Letter 2]

12 Feb 2020

Maternal perceptions of father involvement among refugee and disadvantaged families in Beirut, Lebanon

PONE-D-19-24487R2

Dear Dr. Hein,

We are pleased to inform you that your manuscript has been judged scientifically suitable for publication and will be formally accepted for publication once it complies with all outstanding technical requirements.

With kind regards,

Florian Fischer

Academic Editor

PLOS ONE

---

## [Editor Report · Acceptance letter]

20 Feb 2020

PONE-D-19-24487R2 

Maternal perceptions of father involvement among refugee and disadvantaged families in Beirut, Lebanon 

Dear Dr. Hein:

I am pleased to inform you that your manuscript has been deemed suitable for publication in PLOS ONE. Congratulations! Your manuscript is now with our production department. 

With kind regards,

on behalf of

Dr. Florian Fischer 

Academic Editor

PLOS ONE